# Multiple forest attributes underpin the supply of multiple ecosystem services

María R. Felipe-Lucia, Santiago Soliveres et al.[#]

Trade-offs and synergies in the supply of forest ecosystem services are common but the drivers of these relationships are poorly understood. To guide management that seeks to promote multiple services, we investigated the relationships between 12 stand-level forest attributes, including structure, composition, heterogeneity and plant diversity, plus 4 environmental factors, and proxies for 14 ecosystem services in 150 temperate forest plots. Our results show that forest attributes are the best predictors of most ecosystem services and are also good predictors of several synergies and trade-offs between services. Environmental factors also play an important role, mostly in combination with forest attributes. Our study suggests that managing forests to increase structural heterogeneity, maintain large trees, and canopy gaps would promote the supply of multiple ecosystem services. These results highlight the potential for forest management to encourage multifunctional forests and suggest that a coordinated landscape-scale strategy could help to mitigate trade-offs in human-dominated landscapes.

Forests provide a wide range of ecosystem services, including timber production, carbon (C) storage, local climate regulation and many cultural services associated with recreational activities and nature experience[1–4]. Traditionally, forest management has targeted only a small subset of these benefits, particularly timber production[5,6], which has shaped the vegetation structure and species composition of many of the world's forests. To optimise the production of marketable timber, management typically focusses on growing even-aged, homogenous forest stands dominated by a few economically valuable tree species[7–9]. However, such management can reduce the supply of other ecosystem services, such as carbon sequestration[10], water availability[11] (but see[12]) and aesthetic value[13,14], and can cause biodiversity loss[15]. In some cases, forests are managed for other values like habitat conservation or recreation, at the expense of timber production. However, such management rarely considers a larger number of ecosystem services, and cultural services are often underrepresented[16,17].

In order understand how forest management affects multiple ecosystem services, we need to understand how particular forest attributes, which can be altered by management practices, affect different ecosystem services. Forest management to optimise timber production may decrease tree diversity by favouring a small number of high-yielding and sometimes non-native species. This change in tree species composition, together with more direct effects of management, such as thinning and harvesting operations, alters average tree size, canopy cover and understorey diversity in the stand[18], all of which can affect a range of services such as soil nutrients, C storage[19,20] and cultural services. If forest management reduces the vertical and horizontal heterogeneity of a stand, it may also decrease its aesthetic value[13,21] and reduce pest control (e.g. lead to bark beetle outbreaks[22]), among other services. In addition, forest management alters the amount of deadwood remaining in the stand. In unmanaged forests, deadwood accumulates over time and is a key habitat for many ecosystem service providers[23,24] (e.g. saproxylic insects and saprotrophic fungi driving nutrient and C cycling), while managed forests contain less deadwood[25,26]. All of these forest attributes are likely to be important for different services. Therefore, to get a comprehensive picture of how forest management affects service supply we need to analyse the effect of multiple forest attributes on multiple ecosystem services.

Rather than responding strongly to management, some other ecosystem services may respond more to underlying environmental factors, such as climate, soil depth and slope[27–30]. For example, soil organic C and soil enzymatic activities, both indicators of soil health[31], were better explained by soil texture than by forest management[28,30]. Also, recent studies have shown that soil fungi, the harvesting of which is a popular recreational activity, mostly respond to soil texture and chemical properties[32]. Identifying which services respond more strongly to environmental factors than to forest stand attributes is important as these services are likely to be relatively unresponsive to management.

By affecting services in different ways, forest attributes may also drive synergies and trade-offs between services. For instance, spruce monocultures can increase tree biomass but reduce the yield of wild berries[1], creating a trade-off between these services. In other cases, trade-offs and synergies between ecosystem services might be caused by abiotic or environmental factors[33,34]. For example, increasing temperatures may accelerate the decomposition of organic C and increase N availability[35]. A third possibility is that trade-offs and synergies might be intrinsic, meaning they are not caused by shared responses to environmental drivers or management but are unavoidable due to physiological or ecological reasons[36]. For example, high rates of soil organic matter turnover, driven by decomposer activity, release nutrients but will inevitably reduce soil C stocks[37]. If service relationships are intrinsic or only driven by environmental factors, then changes in management will not be effective in mitigating trade-offs and promoting synergies. Disentangling the interplay between individual forest attributes and environmental factors as drivers of the trade-offs and synergies between ecosystem services is, therefore, fundamental to the management of multifunctional forests. However, although some studies have identified drivers of ecosystem services[36,38], few have quantified the relative importance of different drivers and none have examined relationships between a wide range of different services, forest attributes and environmental conditions.

Here, we investigate the effects of 12 stand-level forest attributes and four environmental factors, on 14 proxies of ecosystem service supply, measured in 150 forest plots. These plots represent common management types and intensities found in Central Europe, namely, managed broadleaved forests dominated by beech (*Fagus sylvatica*) or oak (*Quercus petraea* and *Quercus robur*), managed coniferous forests dominated by to spruce (*Picea abies*) or pine (*Pinus sylvestris*), which includes both even-aged and uneven-aged (selection) systems but no clear-cuts, and unmanaged broadleaved forests dominated by beech[39]. The 12 forest attributes (Supplementary Table 1) represent different aspects of a forest stand which can be altered through management[40] and which we hypothesise could affect ecosystem service supply. They include variables related to stand structure (mean tree diameter at breast height (DBH) and canopy cover), stand species composition (all stands are dominated by conifers, beech or oak and we include only proportion of conifer and oak cover as predictors, as conifer cover is strongly negatively correlated with beech cover, $r = -0.8$), vertical and horizontal stand heterogeneity, tree and shrub diversity (both richness and evenness, note that tree diversity is always low in these forests, i.e. mean richness <5 trees per ha; mean evenness <0.4, see Supplementary Fig. 1), native tree regeneration (regeneration of the dominant native tree species *F. sylvatica*) and deadwood amount (deadwood volume). As environmental variables we considered soil depth, soil pH, slope and location, the latter representing a mixture of local conditions including soil type, climate and elevation (see details in Methods). The 14 ecosystem service proxies include services from the main categories identified by the Millennium Ecosystem Assessment[41]: five supporting services (or ecosystem functions) related to nutrient cycling (root decomposition, dung removal, mycorrhizal diversity, soil N and P availability); four regulating services related to local and global climate regulation (plot-level temperature regulation and C storage in soil and trees) and pest (bark beetle) control; four cultural services related to recreational activities and educational opportunities (edible fungi, wild edible plants, plants of cultural value and bird-watching potential) and, finally, timber production as a provisioning service (Supplementary Table 2). First, we compare the explanatory power of forest attributes and environmental factors in predicting individual ecosystem service proxies (henceforth simply called ecosystem services). Second, we investigate the importance of forest attributes and environmental factors as potential drivers of synergies and trade-offs between ecosystem services. Third, we examine the effects of each forest attribute on individual ecosystem services, to identify attributes with generally positive, generally negative or contrasting effects on ecosystem services.

Our work provides a comprehensive view of the effect of a large range of stand-level forest attributes, all of which are affected by forest management, on multiple ecosystem services. We show that forest attributes are generally stronger predictors of most ecosystem services than environmental factors, although some services respond to both forest attributes and

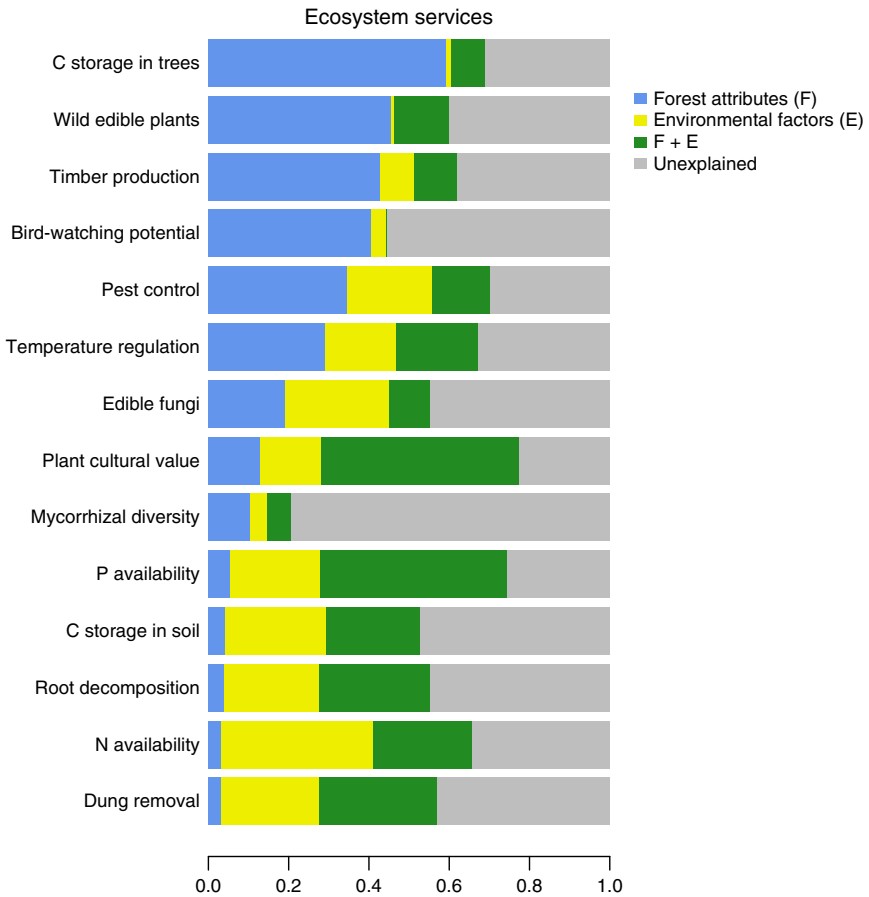

**Fig. 1** Importance of forest attributes and environmental factors for explaining ecosystem services. Bars show the variance exclusively explained by forest attributes and environmental factors, the shared variance between them and the unexplained variance. See Supplementary Table 2 for sample size in each ecosystem service model and Supplementary Fig. 3 for correlation among all drivers

environmental factors. While the effects of forest attributes differed for each particular service, we identified four forest attributes with a positive net effect on services (vertical heterogeneity, understorey richness, mean DBH and conifer cover). Our study highlights the potential of forest management to support forest multifunctionality. Since no individual forest attribute was able to maximize the supply of all ecosystems services, our study suggests that a coordinated landscape-scale strategy is required to mitigate trade-offs and promote multiple ecosystems services simultaneously in human-dominated landscapes.

## Results

**Variance in ecosystem service supply**. Forest attributes were the strongest predictors of most ecosystem services, although some responded to both forest attributes and environmental factors. We used variance partitioning to determine the amount of variance in ecosystem service supply explained by the 12 stand-level forest attributes and four environmental factors (location, soil depth, pH and slope) (Fig. 1). Our models were able to explain on average 59% [21–77%] of the variation in ecosystem services. Forest attributes exclusively explained 23% on average [3–59%] of the variance in ecosystem services compared with 17% [1–38%] for the environmental factors. The shared variance between both groups of variables was usually considerable, explaining on average 20% [0–49%] of the variance in ecosystem services (Supplementary Table 3a). This high shared variance could be caused by correlations between forest attributes and

environmental factors or by other unmeasured variables which jointly affect forest attributes and environment factors (e.g. local hydrology). The five services that were most poorly explained by forest attributes were all soil variables, related to C and nutrient cycling. However, even amongst these, the shared variance between environment and forest attributes was large and only in the case of N availability was most variance explained by environmental factors alone. In contrast, for six ecosystem services forest attributes explained the most variance: these included two cultural services (wild edible plants and bird-watching potential), three regulating services (tree C storage, pest control and temperature regulation) and timber production. Accounting for the different number of predictors included in each model by using adjusted $R^2$ values for the variance partitioning analyses produced very similar results (see Supplementary Table 3b).

**Synergies and trade-offs between ecosystem services**. Correlations between pairs of ecosystem services ranged from −0.60 to 0.51 (Figs. 2 and 3a). When we sequentially removed the effects of different potential predictors (i.e. environmental factors [Fig. 3b], and forest attributes [Fig. 3c]), the strength of the correlations between ecosystem services significantly decreased, both for synergies (positive correlations) and trade-offs (negative correlations). We removed the effect of the environmental factors first, assuming that the environment partially determines forest management and thus the forest attributes, and then we removed the combined effect of environment and the forest attributes, each

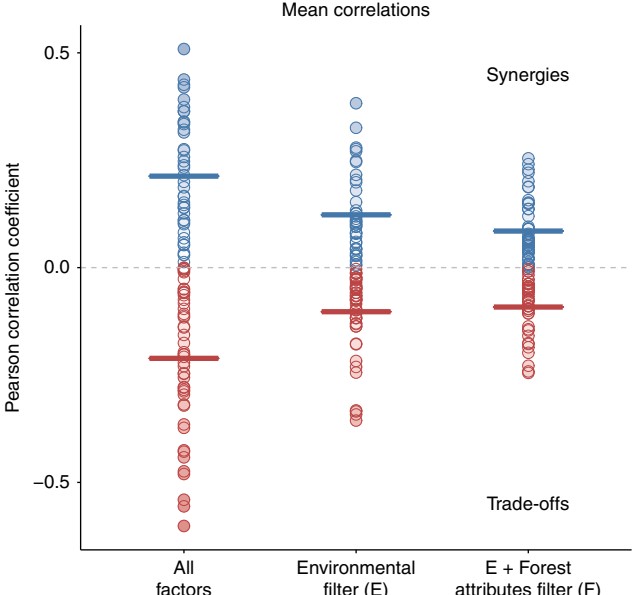

**Fig. 2** Change in the synergies and trade-offs between ecosystem services. Pearson correlations between pairs of ecosystem services (All factors), and after removing the effect of environmental factors (E) and of both the environmental factors and forest attributes together (F). Horizontal lines show the mean positive (blue lines, synergies) and negative (red lines, trade-offs) correlations. See Fig. 3 for details

time recalculating the correlations. This approach highlighted the significant effect that both environmental factors and forest attributes have in explaining synergies and trade-offs between ecosystem services (Supplementary Fig. 4).

Over all services, we did not find a significant, unique effect of the forest attributes on ecosystem service correlations (Supplementary Table 4a). That is, the strength of correlations did not decrease further when the effects of forest attributes were removed in addition to environmental effects, probably because most of the attribute effects are shared with the environment (as shown also in the variance partitioning analysis, Fig. 1). We further investigated if this lack of unique effect was driven by particular services and tested this by dropping sets of services (i.e. by excluding the supporting, regulating, provisioning or cultural services, one group at a time), each time recalculating the correlations and testing for significant differences (Supplementary Table 4b, Supplementary Data 2). We found that when supporting services were excluded, the effect of the forest attributes on the correlations between services became significant. This was not an artefact of including fewer services, as dropping the same number of randomly selected services did not lead to significant differences in the correlations (Supplementary Table 4b). These results suggest that synergies and trade-offs involving supporting services are mainly explained by environmental factors while correlations between the other services can be explained by both environmental factors and forest attributes.

In addition, we found four consistent (intrinsic) trade-offs and four consistent synergies across all correlation analyses (underlined in Fig. 3a). All other correlations depended on different drivers (boxes in Fig. 3b, c). Of the 66 main correlations (those with $r \geq |0.1|$), 28 were affected by the environment (i.e. they changed significantly when effects of the environmental factors were removed) and 11 were driven by the forest attributes (i.e. they were significantly different when forest attribute effects were removed in addition to environmental effects). For example, we

found a significant effect of the environmental factors on the trade-offs between pest control and both N availability and soil C storage, and of the forest attributes on the synergy between tree C storage and temperature regulation (Supplementary Data 2).

**Effects of forest attributes on ecosystem services.** First, we estimated the effects of individual forest attributes on all ecosystem services together, to identify those attributes with net positive or negative effects on ecosystem services. This showed that vertical heterogeneity, shrub species richness, conifer cover and mean DBH increased ecosystem services on average (mixed-model analysis, Table 1). Next, we analysed each ecosystem service separately to identify the effects of individual forest attributes on particular services (Supplementary Tables 6 and 8, and Supplementary Data 3). This revealed that most forest attributes were related to only a small number of services. However, all of them were important for at least one service (Fig. 4; although in the case of deadwood volume and shrub evenness, effects should be treated with caution as they were not significant in the full models, see Methods). Canopy cover, and both conifer and oak cover tended to have contrasting effects on ecosystem services. Increasing values of canopy cover were associated with larger values of local temperature regulation and tree C storage but also with smaller values of pest control, wild edible plants and timber production. Conifer cover had opposing effects to canopy cover and in addition, was negatively associated with the abundance of plants of cultural value and edible fungi. The positive net effect of conifer cover was mainly driven by spruce, not pine (Supplementary Table 7a), and was related to its strong positive effect on timber production (dropping timber from the list of ecosystem services removed this effect). In order to investigate the effects of conifer cover further, we analysed pine or spruce cover in the models instead of overall conifer cover. This showed that pest control, tree C storage and edible fungi were mainly affected by pines, while dung removal and timber production were affected by spruce cover (Supplementary Table 7a). Oak cover also had large and contrasting effects on ecosystem services, as it was related to increased P availability, timber production and bird-watching potential but to reduced local temperature regulation, pest control and abundance of plants of cultural interest. We also investigated potential differences between oak and beech cover in a model that contained these two variables alongside the other forest attributes, but only for those plots which were not dominated by conifers. Most ecosystem services responded similarly to the two broadleaved tree species, but there were species-specific effects on some regulating and cultural services (Supplementary Table 7b). As many of the wild edible plants are also shrubs, we repeated the analysis of this service, excluding the overlapping species, which led to the same results (Supplementary Tables 7c, d). Altogether, these results indicate which of the forest attributes are likely to be most important for explaining synergies and trade-offs between ecosystem services.

**Discussion**
Our results demonstrate the importance of considering multiple forest attributes to understand the drivers of ecosystem service supply[4,42]. Most of our service proxies were more strongly related to the forest attributes alone or to the combined effect of attributes and environmental factors than to the environment alone (Fig. 1). However, the belowground services were largely affected by the shared effects of forest attributes and environment. These services might therefore be less amenable to management because environmental factors are more difficult to modify. In addition, many belowground services are more dependent on slow soil processes (e.g. soil C sequestration) and hence their supply can

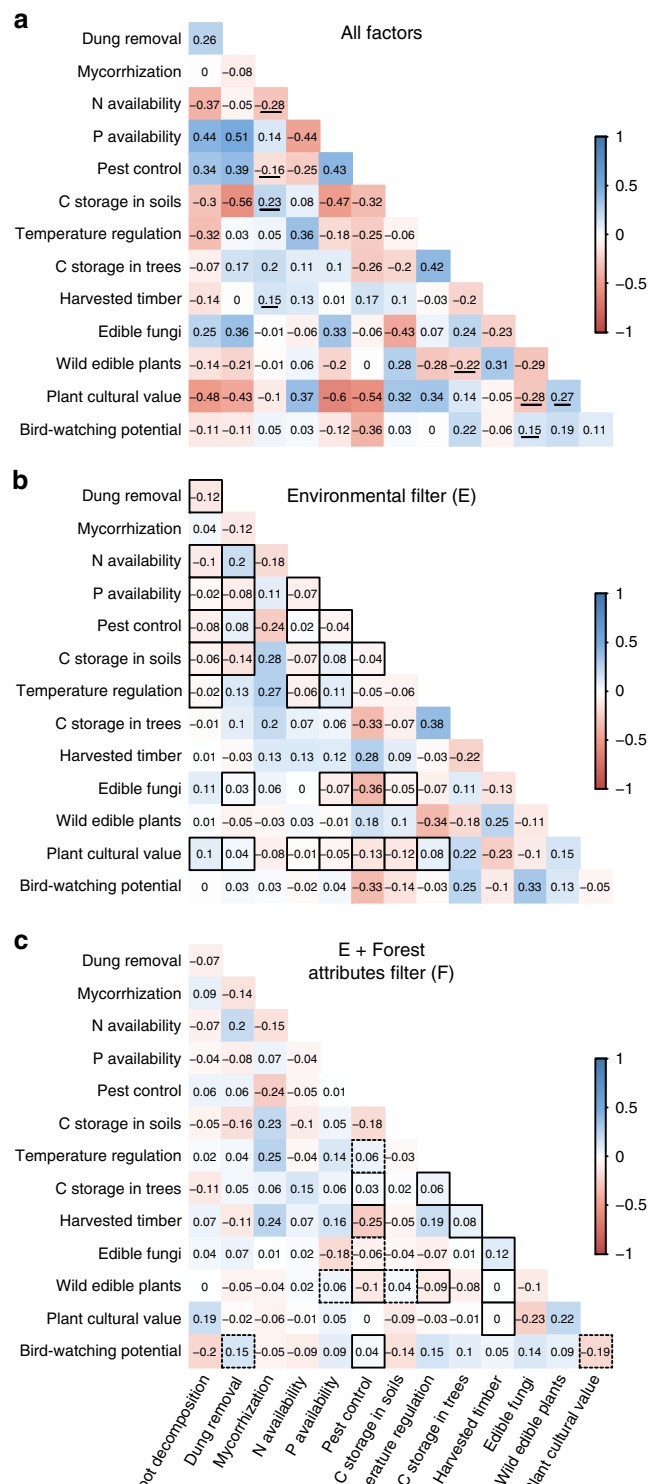

**Fig. 3** Details on the synergies and trade-offs between ecosystem services. Panel **a** shows the individual Pearson correlations between pairs of ecosystem services. Consistent ($r \geq |0.1|$) synergies and trade-offs across the three correlation matrices (All factors, E and F) are underlined. Panel **b** shows the correlations after removing the effect of environmental factors (E). Those correlations driven by environmental factors (i.e. those significantly different from the original correlations, panel **a** vs. panel **b**) are in solid line boxes. Panel **c** shows the correlations after removing the effect of both environmental factors and forest attributes (F). Those correlations driven by the forest attributes alone (i.e. significantly different from the correlation with only environment effects removed, panel **b** vs. panel **c**) are in solid line boxes, and those driven by a combination of the environmental factors and forest attributes (i.e. significantly different from the original correlations (panel **b** vs. panel **d**) but not from those with environmental effects removed (panel **c** vs. panel **d**)) are in dashed line boxes. See Supplementary Fig. 4 for the changes in correlations caused by the environmental factors and the forest attributes separately

shrub richness do not always occur together within management types, which implies that the supply of many aboveground services could be increased in old-growth forests by promoting high structural heterogeneity and diverse understories, as has been suggested previously[25]. Clearly, forest management needs to focus on particular attributes because forests assigned to general management categories, such as managed vs. unmanaged forests, may vary considerably in certain forest attributes (see Supplementary Fig. 1), some of which may have contrasting effects on different ecosystem services (Supplementary Fig. 2). For example, richness of edible fungi increases with mean DBH (Fig. 4); however, mean DBH is not strongly related to management categories and varies substantially within managed and unmanaged forests (Supplementary Fig. 1). These results suggest that particular forest attributes could be targeted to optimise the delivery of many cultural, provisioning and regulating services.

Forest attributes also explained synergies and trade-offs between ecosystem services. Correlations involving supporting services were mainly explained by environmental factors but forest attributes were important in explaining the synergies between the other services (Supplementary Table 4b). For instance, the synergy between timber production and wild edible plants can be explained by the presence of canopy gaps, which are typical in frequently disturbed, managed conifer stands. We also identified consistent trade-offs between ecosystem services that seemed neither driven by shared responses to environmental factors nor by shared responses to forest attributes (Fig. 3). This might suggest intrinsic trade-offs between some ecosystem services, i.e. trade-offs that arise from direct interactions between services, rather than from shared responses to environmental or management drivers. However, we cannot rule out shared responses to unmeasured factors as the cause of these correlations. An example of a likely intrinsic trade-off between ecosystem services, which remained consistent after removing the effects of environmental factors and forest attributes, is the one between N availability and the diversity of mycorrhizal fungi. This trade-off is probably caused by shifts to species-poor nitrophilic mycorrhizal communities in fertile conditions[43–45]. However, for other potential intrinsic trade-offs, the biological mechanism is not yet known.

In general, understanding the drivers of ecosystem service trade-offs and synergies is fundamental for land management decisions and policies[33,36] but tools to identify these drivers have rarely been developed. Our step-wise approach, removing the

only be modified over long time scales. In contrast, the supply of aboveground services like timber production, temperature regulation, C storage in trees and many cultural services could be increased via specific management measures.

Our results clearly show that certain forest attributes, especially mean DBH, vertical heterogeneity and shrub richness had, on average, positive effects on ecosystem services (Table 1). Shrub richness was negatively related to temperature regulation but when all services were taken together its effects were significantly positive. High levels of mean DBH, vertical heterogeneity and

**Table 1 Estimated net effects of forest attributes on ecosystem services**

| Predictor | Estimate | Std. error | DF | t Value | Pr(>|t|) | Significance |
|---|---|---|---|---|---|---|
| Intercept | −0.001 | 0.017 | 1995 | −0.032 | 0.974 | |
| Shrub richness | 0.063 | 0.023 | 1995 | 2.727 | 0.006 | ** |
| Conifer cover | 0.050 | 0.023 | 1995 | 2.170 | 0.030 | * |
| Mean DBH | 0.064 | 0.022 | 1995 | 2.949 | 0.003 | ** |
| Vertical heterogeneity | 0.091 | 0.021 | 1995 | 4.276 | 0.000 | *** |

Four out of 12 forest attributes had a significant effect in a mixed model analysis after model simplification. See Supplementary Table 5 for full models
DF degrees of freedom, DBH diameter at breast height
Significance levels: ***$p \leq 0.001$; **$p \leq 0.01$; *$p \leq 0.05$

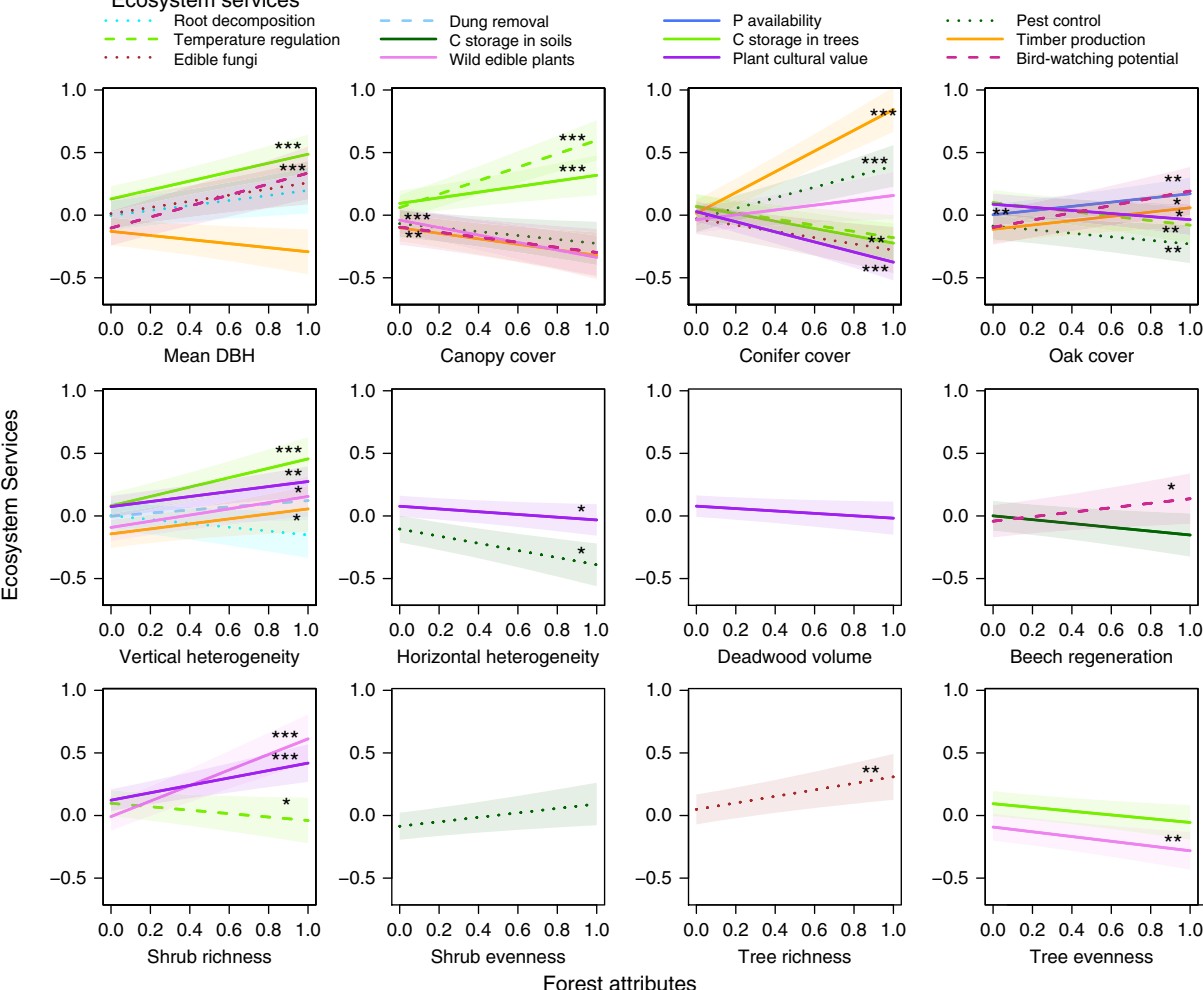

**Fig. 4** Effects of forest attributes on ecosystem services. Standardized partial effects from linear models after model simplification are shown. Shaded lines indicate 95% confidence intervals. Asterisks indicate significant effects also found in the full models (significance levels: ***$p \leq 0.001$; **$p \leq 0.01$; *$p \leq 0.05$, as measured by Student's $t$-test). DBH diameter at breast height. See Supplementary Tables 6 and 8 for details, and Supplementary Data 3

effect of one potential driver at a time, may facilitate a better understanding of the drivers of ecosystem service trade-offs because it allows the effects of different drivers to be distinguished. Although intrinsic trade-offs might be difficult to avoid or mitigate, synergies due to shared responses to forest attributes can be promoted through management practices. Our approach can thus be used to identify key forest attributes that can be managed to promote the supply of multiple ecosystem services.

Our results also show the contrasting effects of many forest attributes on different services (Fig. 4). In general, canopy cover, together with tree species composition (conifer cover and oak cover), were the main attributes that had contrasting effects on services, which indicates that certain services are promoted in unmanaged beech forests with closed canopies, while others are promoted in more open conifer or oak-dominated stands. This is true even for the cultural services, where edible fungi, plants of cultural value and bird-watching opportunities were promoted by beech or oak stands, while wild edible plants were promoted in

open canopy stands (note that conifer and beech cover are negatively correlated, with low values of conifer cover indicating high beech cover). Many studies have suggested that multiple cultural services could be enhanced at the same time[33,46]; however, most of these studies focused on the landscape scale, rather than on the stand level, as we do here. Management could potentially increase the supply of multiple services at the stand level, for example by introducing disturbances in beech forests to create gaps in the canopy.

However, forest management planned at larger scales might be better able to mitigate ecosystem-service trade-offs, if a diversity of forest attributes is maintained at a large scale[47]. For example, it has been suggested that a landscape composed of forest stands varying in tree species identity and diversity would maximise the delivery of ecosystem services at the landscape scale[1,48]. Land-zoning allocation[49], such as the TRIAD approach[50], is another example of landscape scale management that can contribute to landscape-scale multifunctionality by combining unmanaged forests with a mix of low and high intensity forestry. Our results also provide further evidence that variation in species composition (conifer and broadleaf, oak and beech) across a landscape could promote large-scale multifunctionality[48], whilst additionally suggesting that variation in other forest attributes like canopy cover could be similarly important. However, to upscale our stand-level results to the landscape we would need to account for interactions between forest stands (e.g. by moving organisms or material flows), temporal forest dynamics and effects of forest size, connectivity and adjacent land-use types[51]. Additionally, in old-growth natural forests (which are lacking from our study) there might already be sufficient heterogeneity in stand structure at the landscape scale and therefore management might be less important in creating the heterogeneity that may be necessary for high landscape multifunctionality[2,52].

We also found that two measures of forest diversity (vertical heterogeneity and shrub richness) had positive net effects on ecosystem services. These forest attributes were particularly important drivers of cultural and regulating services. Several previous studies have shown that higher tree diversity results in a higher supply of multiple ecosystem services[1,53,54]. However in our study, tree richness only increased one ecosystem service (edible fungi). This may be explained by the fact that in our plots, as in many Central European forests, stands with higher tree species richness are strongly dominated by a single species, often European beech, and include only a few individuals of admixed species (see tree richness and tree evenness in Supplementary Fig. 1). Furthermore, some services might be promoted in pure stands as species may 'specialise' in supplying certain functions and services[55], thus explaining the negative effect of tree evenness on wild edible plants and some effect on tree C storage. Our findings therefore partially support other studies that found relatively weak effects of tree diversity on a range of ecosystem services in European forests[55]. However, the positive effects of heterogeneity and understorey richness suggest that other aspects of forest diversity are important for service supply and these factors might also have contributed to some of the positive effects of tree diversity seen in other studies[1,48,56]. For instance, forest stands with more tree species may have a higher structural heterogeneity and a more diverse understorey; although understorey–overstorey relationships can be complex and understorey diversity is not always high in stands with high tree diversity[57]. This means that tree diversity could increase service supply via its effects on heterogeneity and shrub richness. Furthermore, this suggests that to understand service supply, biodiversity-functioning studies in forests should not only consider the effects of tree richness, but also other components of forest diversity, including tree dominance, shrub diversity and

stand structural heterogeneity. This may also apply to other ecosystems, where attributes other than species richness, such as total abundance[58], functional structure[59] or spatial pattern[60,61] are major determinants of ecosystem multifunctionality.

Our study opens many possibilities for further research but is not without caveats. The gradient of forest management intensity considered by our study is relatively limited because intensive plantation forestry with short rotations is rare in Germany, as are forests that have been unmanaged for over 100 years. Thus, our estimates of the effects of forest attributes on the supply of multiple services, and the synergies and trade-offs between them, may be conservative and we might have found even stronger effects if a wider management gradient had been considered. For example, some of the forest attributes expected to be important for ecosystem services, such as deadwood volume, horizontal heterogeneity and beech regeneration were only found to have a small effect. While increasing the length of the management gradient might reveal that forest attributes cause even stronger trade-offs, our results still show the value of determining the effect of individual forest attributes on ecosystem service supply rather than only considering broad management types. Moreover, we acknowledge that our approach is correlational, and that experiments would be needed to address some of the major hypotheses that emerge from the analyses. Therefore, we encourage further studies in other forest types or management regimes not covered by our study, such as intensive production forests where clearcutting is frequent, primeval or long-term unmanaged conifer forests and also experimental studies that manipulate, e.g. forest attributes.

Our results clearly show that forest attributes are good predictors of ecosystem service supply and of synergies and trade-offs between services, thus highlighting the importance of a careful management of forest attributes. Our novel approach to identifying the main drivers of synergies and trade-offs between services enabled us to tease apart the relative importance of different factors, and can be applied to future studies that include a larger range of drivers. Our findings highlight the importance of wisely managing forest attributes, because of their role in driving most ecosystem services and their synergies and trade-offs. Our study suggests that managing forests to promote closed canopies and old-growth broadleaved forests would boost services related to climate regulation and some cultural services (e.g. edible fungi and plants of cultural value). In contrast, beech and conifer forests with canopy gaps would allow for timber production and other cultural services (e.g. wild edible plants). To inform specific management decisions, forest owners could combine our information on the effects of forest attributes on ecosystem services, with weightings of these services according to specific objectives[62]. Our results suggest that forest management can play a decisive role in promoting synergies among ecosystem services and mitigating trade-offs in human-dominated landscapes. This can be done by targeting specific forest attributes within a stand and by coordinating forest management at larger scales to support multifunctional landscapes.

## Methods

**Study design**. We selected 150 forest plots (100 m × 100 m) in three regions in Germany (50 plots per region) to cover a variety of forest types common in Central European lowland forests. These plots are part of the large-scale and long-term Biodiversity Exploratories program[39] (www.biodiversity-exploratories.de). The south-western region is the UNESCO Biosphere Area Schwäbische Alb; the central region is the National Park Hainich and surroundings and the north-eastern region is the UNESCO Biosphere Reserve Schorfheide-Chorin. The three regions differ substantially in geology, climate and topography, covering a range of almost 3 °C in mean annual temperature and 500–1000 mm in annual precipitation. Within each plot, environmental conditions, such as soil type and slope, are spatially homogeneous to allow the collection of multiple ecosystem service proxies at the same time. Managed plots in our study area include uneven-aged (selection) and even-

aged systems. Even-aged systems employ shelterwood cuts in the regeneration phase. In the past, some conifer plots may have originated from clear-felling, but this practise is not applied anymore. Managed stands undergo intervention every 5–10 years and trees are harvested every 100 years on average. Unmanaged plots have not been harvested for at least 50 years. Our study comprises 25 unmanaged broadleaved forests, 86 managed broadleaved forests and 39 managed conifer forests.

**Ecosystem service proxies**. In each of the 150 forest plots, we assessed proxies for 14 ecosystem services representing the four categories identified in the Millennium Ecosystem Assessment[41]. Supporting services: root decomposition (percentage of root litter mass loss), dung removal (percentage of dung dry mass removed after 48 h), P availability (content of organic Olsen P), N availability (potential nitrification derived from the abundance of nitrifying bacteria) and mycorrhizal diversity (number of mycorrhizal operational taxonomic units). Regulating services: pest control (ratio of the sum of predators and parasitoids vs. bark beetles), temperature regulation (inverse value of the average daily temperature range per plot), C storage in soils (organic C stocks in topsoil) and C storage in trees (C stored in trees' dry biomass). Provisioning services: timber production (mean annual increment across rotation period for even-aged forests and periodic annual increment between two forest inventories for uneven-aged and unmanaged forests). Cultural services: edible fungi (species richness of the edible fungi), wild edible plants (total cover of wild edible plant species known to be collected in the forest), plants of cultural value (plant species of special interest for the general public or for botanists), bird-watching potential (bird species richness). This classification is similar to the more recent Common International Classification of Ecosystem Services (CICES[63]) and the Intergovernmental Platform for Biodiversity and Ecosystem Services (IPBES; which includes ecosystem services in the broader concept of nature's contributions to people[64]) classifications, in which the supporting services are grouped together with the regulating services. See Supplementary Methods for further details.

**Forest attributes**. In each of the 150 forest plots, we measured 12 forest attributes representing unique stand characteristics. The forest attributes included variables related to stand structure, composition, diversity, regeneration and deadwood. Stand structure: mean DBH of the plot, canopy cover (percentage of the plot covered by tree crowns). Stand composition: conifer cover (percentage of total DBH comprised by non-native or planted conifer species), oak cover (percentage of total DBH comprised by all *Quercus* species). Stand heterogeneity: vertical heterogeneity (diversity of canopy layers effectively occupied by foliage and wood from the total number of layers), horizontal heterogeneity (standard deviation of canopy height). Stand diversity: tree richness (number of tree species), tree evenness (evenness of tree species), shrub richness (number of shrub species), shrub evenness (evenness of shrubs species). Regeneration of native species: cover of beech below 5 m height. Deadwood volume: total deadwood volume, including stumps, downed and standing items. See Supplementary Methods for further details and below for the distribution of these variables across forest management types.

**Effect of management on forest attributes**. Our forest plots comprised three broad management types, namely unmanaged broadleaf, managed broadleaf and managed conifer forests. Unmanaged conifer forests are only present in mountainous regions in Germany and do not exist in our sample. In our study, managed forests had lower vertical heterogeneity and tree diversity than unmanaged ones. In particular, managed conifer forests had slightly lower mean DBH and tree evenness, but slightly higher shrub evenness than broadleaf forests. Managed broadleaved forests had the lowest canopy cover, the highest rate of regeneration of native trees (beech regeneration) and intermediate values for the other forest attributes. In contrast, unmanaged broadleaved forests had slightly higher vertical heterogeneity, canopy cover and tree richness than managed forests (Supplementary Fig. 1).

**Environmental factors**. Environmental factors included region (three levels), slope, soil depth (i.e. thickness of the mineral soil layer) and soil pH. These factors might directly influence the supply of ecosystem services, or do so indirectly through forest management. See Supplementary Methods for further details.

**Statistical analysis**. All analyses were performed in R version 3.4.0[65]. To ensure normal error distributions and homogenous variance we first transformed some of the ecosystem service values (log-transformation: wild edible plants; square root-transformation: P availability, pest control, edible fungi, plants of cultural value). We then standardized all response and explanatory variables before the analyses using z-scores (i.e. scaling data to mean 0, standard deviation (sd) 1), to allow us to compare coefficients between forest attributes and ecosystem services. However, this means we may inflate effect sizes for services which have a small range in values (relative to their maximum potential range) compared to services where we sampled a larger fraction of the maximum potential range across our plots.

**Variance partitioning analysis**. We conducted a variance partitioning analysis to compare the variance in ecosystem services explained by forest attributes and environmental factors. Using $R^2$ values from linear models we calculated the variance exclusively explained by both groups of variables: four environmental factors (location, slope, pH and soil depth) and 12 forest attributes (mean DBH, canopy cover, conifer cover, oak cover, vertical and horizontal heterogeneity, tree richness and evenness, shrub richness and evenness, beech regeneration and deadwood volume) and the shared variances between environment and attributes. Shared variance arises due to correlations between environmental factors and forest attributes, e.g. because the attributes are influenced by the environment directly or because foresters adjust their management depending on the environmental conditions. As $R^2$ values could be affected by the different number of predictors included in each model[66], we also conducted variance partitioning using the adjusted $R^2$ (see Supplementary Table 3b), which gave very similar results. For completeness of the results, we also provide all effects and AIC values adjusted for small samples (AICc) of each model used for variance partitioning and a comparison against a null model (i.e. a model without predictors) (Supplementary Table 3c, Supplementary Data 1a-c).

**Analysis of ecosystem-service trade-offs and synergies**. We disentangled the effects of environmental factors and forest attributes on trade-offs and synergies between ecosystem services in a three-step procedure. (1) We first calculated Pearson correlations between standardized ecosystem service data, then (2) we removed the effect of the environment by taking residuals and recalculated the correlations and finally, (3) we removed the effect of the forest attributes and recalculated the correlations. We assumed a hierarchy of controls where the environment partially determines the management shaping the forest attributes, or the environment directly determines the forest attributes. This approach is somewhat conservative in assessing the impact of forest attributes because we assume that they cannot affect the environment and attribute any shared variation between environment and forest attributes to environmental factors.

In step 2, to remove the effect of the environmental factors (i.e. location, slope, soil depth and soil pH) on each ecosystem service, we calculated residuals for each service from a linear model with the environmental factors as predictors. We then calculated pairwise Pearson correlations between ecosystem services, which represent the relationships between the services having removed the influence of shared responses to the environment. We did the same in step 3 to remove the effect of forest attributes by fitting models with all variables (four environmental factors and 12 forest attributes) as predictors, and using the residuals to calculate the Pearson correlation between ecosystem services. We tested for significant differences among the three correlations matrices using series of matrix comparison tests with the R package lavaan[67]. We used the Comparative Fit Index (CFI) and the Tucker–Lewis Index (TLI; also called non-normed fit index, NNFI) to compare the matrices and we used the function lavTestLRT to obtain the unscaled Chi-square ($\chi^2$) value for the difference between matrices. In addition, we computed the McDonald Non-Centrality Index (Mc or NCI). Inspecting all these four indices is recommended to determine differences between correlation matrices[68]. CFI and TLI values above 0.95 and Mc values close to 0.9 indicate that the matrices do not differ from each other[69], while the Chi-square provides a test of whether the matrices significantly differ from each other[68]. We also tested for significant differences between each pair of ecosystem services, following removal of the effects of environmental factors and of forest attributes, using the package cocor[70]. This allows us to identify which individual synergies and trade-offs are driven by shared responses to the environment (by comparing correlations from step 1 with those generated in step 2) or shared responses to forest attributes (comparing step 2 with step 3). We used Zou's confidence intervals with alpha 0.05 (using alpha 0.01 gave the same results) as they are suggested to be superior to significance testing[70]. If the confidence interval includes zero, the null hypothesis that the two correlations are equal must be retained. If the confidence interval does not include zero the two correlations are considered significantly different.

**Forest attributes as predictors of ecosystem services**. We analysed the effect of each forest attribute on individual ecosystem services. Prior to further analyses, we visually inspected all service per attribute relationships and did not find evidence of non-linear relationships. To remove the effect of environmental variables on both forest attributes and ecosystem services we analysed residuals. We first fitted linear models with only the environmental variables as predictors (location, soil depth, soil pH and slope) and with each ecosystem service and forest attribute as the response variable. Residuals from these models were then used in subsequent analyses. Again, this is a conservative approach, because we assessed only the unique effect of the forest attributes on the services and removed any shared variance with environmental factors.

We first tested for overall or net effects of each forest attribute across all ecosystem services. To do so, we fitted one mixed model with all ecosystem services combined, i.e. each service was standardised (z-transformed) and each plot contained up to 14 values for all services measured on that plot (see a similar approach in ref. [58]). Models were fitted using the lme4[71] package. As fixed factors we included all forest attributes and as random effects we included ecosystem service identity and plot. We performed a backward simplification and fitted the simplified model using the lmerTest[72] package based on the Satterthwaite

approximation for degrees of freedom. Effects found in simplified models (Table 1) were similar to those found in full models (Supplementary Table 5).

We next fitted linear models separately for each ecosystem service, to identify effects of forest attributes on individual ecosystem services. We used both backward and forward stepwise model simplification, based on AIC to find a minimal adequate model for each response, using the function stepAIC (MASS package[73]). We then further simplified the output models to remove all non-significant terms using $F$-ratio tests, comparing models with and without the term of interest. We plotted the significant effects of each predictor on each ecosystem service using the visreg package, which shows predicted slopes and confidence intervals around the predictions[74].

As model simplification results in multiple comparisons, we also tested the significance of each forest attribute in full models, which were not simplified. A total of 42 effects were significant in the simplified models and of those 29 were also significant in the full models (three additional variables were significant only in full models but we do not consider them further). To further account for the fact that including many predictors in the models could increase the chances of type I error (false positives), we also estimated adjusted $p$-values for the terms in these full models following the Benjamini–Hochberg procedure[75] to detect false discovery rates (FDR). Considering a very conservative approach of FDR = 5%, we found 11 significant effects (marked with three asterisks in Fig. 4). Considering a more commonly used FDR = 25%, 17 additional effects were significant (marked with one or two asterisks in Fig. 4). As our analysis was partially exploratory, we show all 42 effects, to provide hypotheses for further studies. However, caution should be taken in interpreting the results of the effects not significant in full models. In addition, we provide the unscaled relationships between forest attributes and ecosystem services for those effects found significant in the simplified models (Supplementary Fig. 5).

We further tested the effect of conifer species identity by replacing conifer cover by pine cover or by spruce cover and re-running the analyses. We also tested potential differences between oak and beech cover fitting these two variables together with the rest of forest attributes, we conducted this analysis only on the broadleaved dominated stands. Because 27% of the shrub species are wild edible plants, we also estimated shrub richness and evenness excluding the overlapping species and re-ran the analyses, which did not change the results.

## Data availability
This work is based on data elaborated by several projects of the Biodiversity Exploratories program. Part of the data used are available at https://www.bexis.uni-jena.de/PublicData/PublicData.aspx (IDs: 1000; 6241; 15386; 16506; 17687; 17706; 18271; 20035; 20106; 20366). However, to respect the rights of PhD students of individual projects, the data and publication policy of the Biodiversity Exploratories includes an embargo period of 3 years for the remaining data (IDs: 11603; 14447; 14686; 16666; 17086; 19007; 19186; 19286; 19847; 19866; 19986; 20607; 21047; 21449; 22868), which will be made publicly available at the same website.

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

## Acknowledgements

This is a product of the Synthesis Core Project of the Biodiversity Exploratories. We thank the managers of the three Exploratories, Kirsten Reichel-Jung, Katrin Hartwich, Sonja Gockel, Kerstin Wiesner and Martin Gorke for their work in maintaining the plot and project infrastructure; Christiane Fischer and Simone Pfeiffer for giving support through the central office; Michael Owonibi for managing the central data base; and Eduard Linsenmair, Dominik Hessenmöller and the late Elisabeth Kalko for their role in setting up the Biodiversity Exploratories program. The work has been supported by the DFG Priority Program 1374 "Infrastructure-Biodiversity-Exploratories". Field work permits were issued by the responsible state environmental offices of Baden-Württemberg, Thüringen and Brandenburg (according to § 72 BbgNatSchG). S.S. was supported by the Spanish Government under a Ramón y Cajal contract (RYC-2016-20604).

## Author contributions

M.R.F.-L. and E.A. developed the ideas for this manuscript and with S.S. defined the final analyses. M.F. initiated the Biodiversity Exploratories project aimed at measuring multiple ecosystem services in the forest sites. M.R.F.-L. analysed the data and wrote the first draft of the paper. E.A., S.S., C.P., P.M., F.v.d.P., S. Boch, D.P., C.A., P.S., M.M.G. and M.F. substantially contributed to the revisions of the manuscript. A.d.F. assisted in the analyses and graphical outputs. S. Boch, D.P., C.A., P.S., M.M.G., J.B., F.B., S. Blaser, N.B., M.E., K.F., K.G., F.H., K.J., T.K., T.N., Y.O., R.P., A.P., S.R., M. Schloter, I.S., M. Schrumpf, E.-D.S., E. Solly, E. Sorkau, B.S., M.T., W.W.W., T.W., M.F. provided data. All authors commented on the manuscript.

## Additional information

**Competing interests:** The authors declare no competing interests.

María R. Felipe-Lucia [1], Santiago Soliveres[1,2], Caterina Penone [1], Peter Manning [3], Fons van der Plas [3,4], Steffen Boch [1,5], Daniel Prati[1], Christian Ammer [6], Peter Schall[6], Martin M. Gossner[5,7], Jürgen Bauhus [8], Francois Buscot[9,10], Stefan Blaser[1,5], Nico Blüthgen[11], Angel de Frutos[1], Martin Ehrbrecht[6], Kevin Frank [11], Kezia Goldmann [9], Falk Hänsel[12], Kirsten Jung[13], Tiemo Kahl[14], Thomas Nauss[12], Yvonne Oelmann [15], Rodica Pena[16], Andrea Polle [16], Swen Renner [17], Michael Schloter[18,19], Ingo Schöning[20], Marion Schrumpf[20], Ernst-Detlef Schulze[20], Emily Solly [5,20], Elisabeth Sorkau[15], Barbara Stempfhuber[18], Marco Tschapka[13,21], Wolfgang W. Weisser[7], Tesfaye Wubet [9,10], Markus Fischer[1,3] & Eric Allan[1]

[1]Institute of Plant Sciences, University of Bern, Altenbergrain 21, 3013 Bern, Switzerland. [2]Department of Ecology, University of Alicante, Carretera de San Vicente del Raspeig s/n, San Vicente del Raspeig, 03690 Alicante, Spain. [3]Senckenberg Biodiversity and Climate Research Centre (SBIK-F), Georg-Voigt-Straße 14-16, 60325 Frankfurt, Germany. [4]Department of Systematic Botany and Functional Biodiversity, University of Leipzig, Johannisallee 21-23, 04103 Leipzig, Germany. [5]Swiss Federal Research Institute for Forest, Snow and Landscape Research WSL, Zürcherstrasse 111, 8903 Birmensdorf, Switzerland. [6]Silviculture and Forest Ecology of the Temperate Zones, University of Göttingen, Büsgenweg 1, 37077 Göttingen, Germany. [7]Department of Ecology and Ecosystem Management, TUM School of Life Sciences Weihenstephan, Technical University of Munich, Hans-Carl-von-Carlowitz-Platz 2, 85350 Freising, Germany. [8]Chair of Silviculture, Faculty of Environment and Natural Resources, University of Freiburg, Tennenbacherstr. 4, 79106 Freiburg, Germany. [9]Soil Ecology Department, Helmholtz Centre for Environmental Research – UFZ, Theodor-Lieser-Straße 4, 06120 Halle (Saale), Germany. [10]German Centre for Integrative Biodiversity Research (iDiv) Halle-Jena-Leipzig, Deutscher Platz 5e, 04103 Leipzig, Germany. [11]Ecological Networks, Department of Biology, Technische Universität Darmstadt, Schnittspahnstr. 3, 64287 Darmstadt, Germany. [12]Environmental Informatics, Faculty of Geography, Philipps-University Marburg, Deutschhausstr. 12, 35037 Marburg, Germany. [13]Institute of Evolutionary Ecology and Conservation Genomics, University of Ulm, Albert-Einstein Allee 11, 89069 Ulm, Germany. [14]UNESCO Biosphere Reserve Thuringian Forest, Brunnenstraße 1, 98711 Schmiedefeld am Rennsteig, Germany. [15]Geoecology, University of Tübingen, Rümelinstr. 19-23, 72070 Tübingen, Germany. [16]Forest Botany and Tree Physiology, University of Goettingen, Büsgenweg 2, 37077 Göttingen, Germany. [17]Institute of Zoology, University of Natural Resources and Life Sciences, Gregor-Mendel-Straße 33, 1180 Vienna, Austria. [18]Research Unit for Comparative Microbiome Analysis, Helmholtz Zentrum München, Ingolstädter Landstr. 1, 85758 Oberschleissheim, Germany. [19]Technical University of Munich, Emil-Ramann-Str 2, 85354 Freising, Germany. [20]Max Planck Institute for Biogeochemistry, Hans-Knöll-Straße 10, 07745 Jena, Germany. [21]Smithsonian Tropical Research Institute, Luis Clement Avenue, Building 401 Tupper, Balboa Ancón, Panama

