## [Peer Review File · Nature Communications]

Reviewers' Comments:

Reviewer #1:

Remarks to the Author:

Review of Felipe-Lucia et al.

Comments for the authors.

This is a potentially very interesting and novel paper on the drivers of forest ecosystem services, based on a large data set from three extremely well studied German sites, the so-called Biodiversity Exploratories. The data come from ≈ 150 plots in total, and are analysed in novel ways to explore the linkages of multiple ecosystem services and the environmental, management and forest structural variables that may affect the levels of and correlations between the services. It has the potential to advance the field of ecosystem services research substantially, once a number of critical things have been sorted out (see below). The authors also emphasise the management implications of their results; however, this is more contentious and will need substantial re-working and re-thinking before it can be argued with confidence and communicated with forest managers and policy-makers.

I was initially impressed by this work and its attempt to shed light on ecosystem multifunctionality. The statistical analyses seem adequate, sometimes quite elegant, simple to understand and conservative, and addressing the questions well. However, I have a number of concerns, comments, questions and requests for clarifications that need to be addressed before I am able to recommend anything on this MS:

Firstly, I find the choice and classification of ecosystem services (ES) unclear and inconsistent (pt 1 below). Not only are the ecosystem services classified in a haphazard way, some of them are also questionable and cannot be interpreted as proxies for any service or ecosystem function.

Secondly, the broad classification of the factor "management" into 3 classes, unmanaged, managed conifers and managed deciduous forests, is too coarse to be useful and does not reflect the management practices that produce the forest attributes that are argued to be main determinants of ecosystem services. In the text, there are several examples that management actually determine forest attributes, but it is surprising that the consequence is not realised by the authors. This implies that one cannot argue that forest structures/attributes are the drivers, since they are produced by management, and can be changed by management (point 2 below).

Thirdly, dominant tree species is not included among the independent variables that are related to ecosystem services. It is well known that tree species are important "ecosystem engineers" that affect many ecosystem functions (which is also acknowledged in several places in the text). It may be that the dominant tree species is confounded by region in this study. If so, it must be clearly stated, but then the value of the study is in my view diminished (point 3 below).

Fourthly, although the analyses are elegant, I am of the opinion that several of the interpretations cannot be made and that the conclusions are debatable. There are also some statistical issues that I would like to be clarified (point 4 below).

Most if the above points would have been handled, if you had thought a bit more about the system you're examining. Had you made a clear systems diagram, or thought about the questions from a structural equation point of view, you should have realised that production forests, their structure and function, are driven by management choices, and even if you didn't have full information it would have been a latent variable that drives the system. I am NOT advocating a structural equation approach, your questions are different, but you should have thought a bit more what actually produced the forest

attributes you centre your MS on.

I also believe that there is an unfortunate mix of attempted generality and high specificity of the results that diminish the quality of the MS. Its value lies in providing a general framework on how ecosystem services and the relations between multiple ES can be examined, given that there are a number of environmental, ecological, social, management and economic variables that can explain these relationships. But the authors also resort to discussing a number of very specific details when trying to explain the results, at the same time as they omit other details on biology and ecology. That makes the results puzzling and questionable. An example is the discussion on species composition and mixtures, which is even proposed as a management practice but not included in the analyses. I have a large number of more detailed comments (below) that, among other things, clarifies this problem. Any new version of this MS needs to have a better balance between the general and the specific than the present version.

I have spent quite a long time on this MS, since I believe it's a very interesting one, that needs quite some more work.

1. The classification of ecosystem services is not consistent throughout the paper. They start by using the MA classification, viz. provisioning, regulating, supporting and cultural services. This classification is easily collapsed into the CICES one by combining regulating and supporting, so it's not really a problem, just a bit oldfashioned as regulating and supporting services are grading into one another and often very similar. But later on, the authors use "final" ES as services that are "directly used by people", combining provisioning services such as timber production and harvesting with cultural services. This usage is awkward and conflicting with all other classifications that I know of. It is always emphasised that cultural services should be kept distinct as they are fundamentally different than biophysical ones, a notion that is also consistent with the critique on ES from social sciences. The problem is that I believe this greatly affects the interpretation of the results. It is argued that intermediate services (supporting and regulating) are more affected by the environmental conditions, while final (i.e. both provisioning and cultural) are more affected by forest attributes. However, from Figure 1b, it is clear that the "final services" may or may not show this pattern. C storage in trees, harvested timber show this, while temperature regulation does not, and neither does C storage in soil (because it has a large environmental component). Among the cultural services wild edible plants, bird-watching potential (i.e. number of bird species seen) and to some degree edible fungi shows the pattern, while floral aesthetic value and educational value definitely do not follow the pattern. And in Figure 1a it is clear that the two fungal diversity indicators, that are contested proxies of some ecosystem functions, do not have a large environmental component. In order to make the point they want to make, the authors have to combine two disparate classes of ecosystem services that have absolutely nothing in common, and even then the patterns are ambiguous. To propose this is a major outcome of their study, as in the abstract, is simply going too far, in particular since there is no statistics backing up this pattern. The variance partitioning does not produce any generalised statistic, as far as I could find.

In my view, it would be better to stick to the originally chosen MA classification, analyse the ES according to this, and realise that there are some services that are affected by forest attributes (which I interpret as more specific management practices, but that's another point) and some that are not. It is more enlightening to see which these former or latter services are. Or retort from advanced statistics and see if some simple things like rank tests could be illuminating.

Finally, on this point, many of the ES proxies are somewhat stretching the imagination:

To use fungal diversity as proxy for decomposition or pest suppression lacks good scientific evidence. The relations between diversity and processes cannot be assumed, it should have solid evidence behind it.

Temperature regulation is questionable. Do forests really contribute to temperature regulation that

has a societal value? If so, it's related to recreation, or it has a large scale effect on temperatures experienced by people in other ways. I found no references with evidence for this, and it should be discussed a bit more before accepting it as a provisioning service. Or is it cultural?

Harvested timber is somewhat problematic. It does not measure the supply, but rather of demand – the potential supply will be there even if the demand disappears. The argument that timber supply in unmanaged forest is zero is wrong. It's the demand that doesn't exist. In my view, biomass production would be a better proxy for potential timber harvest, i.e. supply, but if this supply is used is another thing. But this is contentious, however. I think you may need a better argument. But in addition, the variable is calculated from the stumps found in the plots. In order to know what this variable means (and hence the time frame of this variable) one needs an estimate of the decomposition rate of tree stumps. Is it 20 or 50 years? This makes a large difference for understanding this proxy. If stumps don't decompose for 50 years, it is a decent proxy of historical harvesting rates, but if stumps disappear in 20 years time, then it's not producing an indicator of the harvesting rate that produced the structures/attributes analysed.

Edible fungi has a list of "edible species", although there is no source for the information. However, many of the "edible species" are of the Amanita genus, which contains many poisonous species and it is generally argued that they should not be picked at all because of the risk of confusing them with highly poisonous ones. One species is redlisted (battarae) and two are likely somewhat poisonous (fulva and rubescens). The other species in the list look acceptable, however. But, how many of them are often and likely to be picked by the public, and is it a provisioning or recreational service?

The plant lists are quite overlapping, and I am a bit surprised that these three services are treated as separate as they partly rely on the same species, but I accept that – although it means that they are not independently estimated. More worrisome is the fact that "floral aesthetic value" is only comprising the spring-flowering species and not, for example, the 6 orchid species in Table S13, nor Fragaria, Hypericum or Prunus from Table S11. It seems to me that not much thought has gone into these proxies, but perhaps the "experts" used differ from me in this respect. But really, these cultural services should have been evaluated together with stakeholders, like amateur botanists, people using forests for recreation, etc.

It is good to try to examine many ES, but the question is how much evidence you need to do it. In my view, some more caution is needed for some of the proxies.

2. The broad classification of the factor "management" into 3 classes, unmanaged, managed conifers and managed deciduous forests, is problematic. It is too broad to be useful, especially since you actually want to make a case about which ES depend on management (well, here you mainly state ES do not depend on management). Of all the ES, only harvested timber is largely explained by management (Fig 1) but that's because unmanaged forest cannot produce any timber (according to the definition of the ES). But these three categories do not adequately reflect the other management practices in these forests. It is management that produces the forest attributes that are argued to be main determinants of ecosystem services. In the text, there are several examples of this:

Carbon storage in trees, for example, is of course directly related to forest management, in the sense that management is determining how much is harvested (not only if, as the three categories suggest) and when (i.e. DBH) the forest is harvested. All ES that depend on forest structure and canopy cover, e.g. temperature regulation, the plant ES, are determined by management, not by the structures that management has produced – i.e. it is harvesting, thinning, and other management interventions that are important, THE ATTRIBUTES ARE ONLY PROXIES OF MANAGEMENT INTERVENTIONS. Also, they come in clusters related to management practices, but it is not clear what these clusters might be since the full data are not included.

So, management actually determines forest attributes, and it is surprising that this is not realised by the authors. One cannot argue that forest structures/attributes are the drivers, since they are produced by management, and can be changed by management, as indicated in the text later on, on e.g. lines 286-289, line 305-312, lines 350-352 where it is argued that open canopies and

disturbances are due to harvesting, lines 396-399, etc. On line 401, it is not realised that tree richness (and tree species composition) is also determined by management in most forests (during thinning, for example). And on line 417 mixed forests are suggested to promote some services – but mixing species is also a management practice not caught by your “management” factor.

It could be that German forests are managed in such a simplified scheme as suggested, but if so – which I doubt – then this study is just a case study of a particular way of managing forests that has limited general relevance. I hope this is not the case, because I believe the German model as well as the Biodiversity Exploratories have a lot to contribute in general.

I needed to explore this in some detail, because it means that your conclusion that forest attributes is more important than “broad forest management types” (line 409) confuses the important issue and makes this study of quite small policy relevance. It confuses (well, hopefully not) policymakers into stating that management is unimportant for forest ES, when in fact it is, and any forest manager knows this. You really need to address this in a more clever way, and start from the obvious fact that management in production forests is a main determinant of forest structure and functioning. The unmanaged forests here are interesting, but irrelevant for production forests, which still is the main proportion of European forests, and something that we can influence.

3. A major drawback, that limits the applicability of the study, is that dominant tree species and species composition were not included among the independent variables that are related to ecosystem services. You only use the less ecologically informative tree species richness. It is well known that tree species are important “ecosystem engineers” that affect many ecosystem functions. There are many studies of the effects of single species like beech, spruce, pine and other species differ in their effects on various ES (see e.g. experimental studies carried out in the 1960-90ies). This fact is also acknowledged in the text.

I could imagine that dominant tree species is confounded by region in this study, but that’s no excuse to omit the probably most important (and managed) variable that produces variation in ES between plots. It could also have been excluded when environmental variables were selected – there is not any information on which variables were discarded and why – but using mathematical/statistical algorithms to exclude the most biologically relevant variables seems to me a bit thoughtless, to say the least, so I hope you didn’t do this.

I must remind the authors that in most biodiversity-ecosystem functioning studies, species composition is more important for ecosystem functions than species richness. So there are strong reasons to use species composition or dominant tree species in your analyses. Since I would guess that there are at least three species – beech, spruce(s) and scots pine (perhaps other pines) – that are dominant, the use of species identity links directly to traits of species and thus to important ecological mechanisms underlying the effects on ES.

I am of the opinion that the analyses need to be redone with dominant tree species and/or species composition as explanatory variables. If not, the authors have to have VERY GOOD arguments to not do it. And not doing it diminishes the value of the study greatly.

4. The points 1-3 above lead me to the conclusion that several of the conclusions are not warranted, and not really supported by the data. It is not correct to state that forest attributes are the main drivers of intermediate ES, because the attributes are largely proxies of past management. The amalgamation of provisioning and cultural services into “direct” services is confusing and inconsistent with the MA classification used and much of the other work on ES. The statistical support for the two conclusions is not convincing, because no real statistical analysis was made of these patterns. The most interesting part of the study is the attempt to find out which variables were the main determinants of ES by using analyses of the residuals from models including and excluding environment, “management” and (only including) attributes. I thought this was an elegant idea, but still there are a number of statistical issues that I need to bring up.

Firstly, since you are dealing with a correlation structure, the correlations in that structure are to some

degree dependent on each other. For example, if A and B are positively correlated, then if C is negatively correlated to A, it is less likely that it's positively correlated to B. So when you test the significance of correlations in Figure 2, these correlations are not independent of each other, and you cannot (without some adjustment or randomisation) use the Tukey HSD test.

Moreover, with respect to Fig. 2, the conclusion that "forest attributes influences the negative correlations between immediate services (implicit: more than environment)" is not warranted. Neither is the statement "attributes were ... main drivers of synergies between final services". While I agree that the pattern for synergies in Fig. 2a suggests that environment is the most important, neither Fig. 2a trade-offs or Fig. 2b synergies show anything else than a gradual decline as more variables are added, they DO NOT show any difference between E+M+F and E+M or E (b is NOT different from a, b). The results do NOT show that the addition of F in E+M+F is different from E alone, so you can't make any inference that the addition of F is more important than E alone, especially in the light of the Tukey tests likely being overestimating differences. This is a VERY elegant way of testing the hypothesis, but it does not support the conclusions, apart from clearly showing that addition of E decreases correlations between synergies.

If I am right, this questions the analyses of forest attributes on final ES reported on line 271 onwards, because here the authors assume that final ES are driven by forest attributes, but as I have argued this is not warranted from neither Fig. 2b nor Fig 1b. While this analysis is interesting, the two Figures rather suggest that environmental variables should be included in a more full analysis of what drives ES.

Furthermore, in Table S5, the authors adjust the P-values for multiple comparisons, but this is not the problem. The authors should have adjusted the P-values for the non-independence of the correlation coefficients before doing anything else. They need to figure out how.

In Fig. 2, the standardised effects are shown, which is OK given the analysis, but it does not show any information on the actual relations in the field, which from a more practical point of view would have been desired. But this is just a small issue.

Similar to this, the use of z-scores is good for standardising the independent variables, but since it makes them equal in the analyses it does not include the fact that they may have very different degrees of variation (as suggested in Fig. S1, where some box plots shows large and some small variation). This means that the major long gradients are downweighted relative to shorter gradients. Does this matter? Perhaps. But for a good understanding and hence providing management and policy advice, it is important to understand this in more detail. You need to consider the possibility that your analyses may overestimate the importance of short environmental gradients.

In the variance partitioning, you are using 11 attributes but only 3 management and 4 environmental factors. I am not a real expert on this, but I have understood that it is important to have similar numbers of variables in the different larger categories when doing the variance partitioning, otherwise you give undue weight to the category with the largest number of variables. You state that you solve this by using adjusted R-squares, but I'd really like you to consult a statistician on this before you make this statement. Or perhaps you already have done, in which case it should be stated in the supplementary materials.

Furthermore – and I guess this is a statistical point to some degree - the most thoughtprovoking and interesting result (despite my criticism of it) is that you can (perhaps, and my interpretation) take the full set of correlations between ES and sequentially decompose it into those determined by the naturally given "environmental" factors, which cannot really be managed on the short time scale, those determined by "management" as such, and those dependent on those structures/attributes that are (my interpretation) created intensionally or unintentionally by management practices. This idea is very clever, but I would not have hidden it in the supplementary material. It is actually the main novelty of the paper (in my view) and deserves more than the ambiguous Figure 2. Can it be shortened and placed in the main text as a Figure or Table? And what do we learn from it? Which ES are synergistic because they are driven by the same environmental drivers, and which are synergistic after these have been taken away, and thus can be influenced by management (since after E+M+F

very few are left), which is what is needed to influence policy. A table or the set of correlation matrices would be a better presentation than the tedious discussion in the text, lacking good visual images or a table, as it is now. I suggest you consider this.

All this boils down to me being very uneasy about the conclusion on lines 408-409 that forest attributes are better predictors of ES than "broad forest management types are". Although one could argue that this is formally correct given the included variables, analyses, and their interpretation (which I am sceptical about, see above), it gives the impression that management is not very important, which is wrong. As argued above most attributes are produced by management, and the statistical support for the conclusion is not very convincing when being scrutinised. Moreover, it should be pointed out although elegant, the approach in this MS is correlational, and that experiments would be needed to address some of the major propositions/hypotheses that emerge from the analyses.

5. Before going into minor details, I want to point out some other problems:

The idea of "intrinsic" trade-offs is confusing and ambiguous. I assume you mean that these are trade-offs that result from basic ecological or biophysical limitations or processes, such as the examples on litter decomposition and C accumulation (line 143) or on mycorrhizae and nitrogen availability (line 251). I think, however, that the term intrinsic is confusing, when it rather should be called something like "unavoidable trade-offs for physiological or ecological reasons".

The environmental variables are presented in the main methods section, but the descriptions of the ES proxies are relegated to the supplementary materials. This is inconsistent, since the paper is about ecosystem services – not environmental variables.

Furthermore, it is unclear which of the original forest attributes were not included in the final analyses, and how they were related to the ones selected. You should, in the supplementary materials, present a list of the original 20 and a correlation matrix that indicates how they related to the ones you finally used. And, as indicated above, I suspect that some of the excluded variables would be more biologically/ecologically relevant – but I may be wrong. You also need to explain better why the diversity measures used are adequate proxies of ecosystem services (see above).

6. Minor details in the text:

Line 84. The term "optimise" is ambiguous. In the case here, the production of marketable timber is rather maximised and for profit but under the constraint of national laws and regulations, so any optimisation is with respect to profit maximisation.

Line 88-90. The examples of trade-offs between intensive timber production and other services are not the best ones. What is it that is sacrificed when intensive production is performed? It is the supporting and regulating (and cultural) services, according to UKNEA and other reports. But you only use examples like water availability (apparently ambiguous), biodiversity loss and cultural services.

Lines 94-102 (approx). The question here is if the variation this study is based on is large enough to address these issues. Many boxes in Fig S1 are fairly narrow which suggests gradients are not long, with the exception of Conifer cover which instead lacks the middle range and seems to be either 80-100% or 0-10% (mainly).

Line 105. Again the term "optimise" emerges from nowhere. What is supposed to be optimised here, and for whom? Landowners, forest industry, the public, or society? Since there are multiple interests in society concerning how forests should be managed, optimisation is always with respect to something. And ecosystem services are often regarded as public goods, so the "optimum" depends on how power over management is distributed between landowners and society.

Line 115-123. Here it is clear that management determines structures and attributes, but then you forget this in the rest of the MS. These examples of management are not caught by your 3-level "management" variable. (this is also relevant for lines 152-160).

Line 161. What does an evenness measure mean, in forests where dominance structure is clearly very uneven (text and e.g. Fig S2 conifer cover panel).

Line 169. For whom and on which scale is temperature regulation interesting? Does it have local,

landscape or even regional impacts?

Line 174. I have already discussed that it's not useful to combine provisioning and cultural ES. Remember this!

Line 194-196. The recommendation on forest management practices suggested here is without substance when the results are scrutinised (also relevant for lines 201-202). It is not clear which management practices you mean, and you are in other places rather arguing that management is not important, so the paper is inconsistent on this issue.

Lines 206-210. The amount of variation explained by the models varied quite much – from 20 to 80% for individual ES. It is evident in Figure 1, but should also be pointed out here.

Further, are there any statistics supporting that forest attributes or environmental factors really explained MORE than the other variables in the two cases? If not, the conclusions should be weaker.

Figure 1. This figure shows the results to be much more variable within your categories than you argue in the text (lines 206-216). So there is a need for a statistical evaluation of the results from the variance partitioning. Note that I suspect (I may be wrong) that there is really no statistical result that backs up the statements you make.

Figure 2. The results are not showing what you state they do, since the Tukey tests are misinterpreted, because the correlations are not independent data points, and because there is mostly no significant differences between correlations when just E is included and when E+M or E+M+F has been accounted for..

Lines 250-254. The "consistent" correlations are sometimes expected, sometimes seemingly haphazard (like saprotrophic diversity vs. dung removal, mycorrhizal diversity vs pest control), and sometime because the ES proxies are using partly the same underlying data (the plant-based ES). But what confidence should we have in these patterns? Are there non-measured variables that could explain them?

Line 280. Are there any possible mechanisms for tree evenness having negative (or positive) effects on ecosystem services? Or what is going on here? Is it a useful variable given that the forests are dominated by mainly one species?

Lines 281-288. The amount of detail here is bewildering and confusing, although many of the relationships look like they are real and not artifacts, and driven by e.g canopy cover and light competition. However, I can't help pointing out that again the authors discuss the effects of harvesting activities, i.e. management, affecting ES.

Figure 3. The panels are too small, and it's not reader-friendly to use exactly the same type of lines – they should be distinguished both by line type (whole, broken, dotted) and colour. What is the 95% CLs referring to? Slope? If so, should it not be larger at both ends.

Line 305-308. The effect of DBH suggests that age of the forest has an effect, but it's not included as a variable. Why?

Line 310-312. Here the authors bring up managing for mixed forests, but there is nothing in the results that suggests anything about mixed species forests. And is the introduction of small-scale disturbances feasible without any ideas about which actual management practices are producing which structures?

Line 324. The synergy between carbon storage and temperature regulation is most likely due to both being related to a third variable. C storage and temperature regulation are acting on entirely different time-scales, so it's highly unlikely there is a direct relationship between them. The closed canopy explanation is not on the time scale of C storage – rather C storage reflects the history of land use during the last 50-100 years.

Line 329-30. The two explanations put forward are both possible, but very different biologically and management-wise. If the result still remains after new analyses you may have to think about this. In general, the discussion is too long and has an imbalance between detail and generality that I mentioned earlier.

Line 355. The cultural services are not driven by the species of interest, but rather by the interests of particular people or stakeholders, for instance those interested in recreation or bird-watching. And

there's not societal interest in these, rather just a stakeholder interest (which is good enough to call them ES, though).

Line 357. *Vaccinium myrtillus* and *Rubus idaeus* are two plants with entirely different biology and traits, and unlikely to be affected by the same management. *V. myrtillus* is more of an old-growth plant, and *R. idaeus* a clear-cut or disturbance-favoured plant, at least at more northern latitudes. Line 358. The "appealing plants" are all to be considered "spring-flowering" which is a bit awkward since many other plants among those in your three lists are to be considered appealing, e.g. the orchids.

Lines 371-376. This part needs to be re-thought after new analyses have been made. The ideas on landscape management are fine but may need re-thinking.

Lines 384-387. These are key lines in my reasoning, but they are unclear on what they mean. It does argue that these forest plots are indeed dominated by one species like beech, and with only few individuals of other species. Hence tree species composition, least if including relative abundances, is likely to be very important, but still it is not included. This is in the light of these lines a bit surprising. And despite the problem written out with the tree diversity variable, it is retained. I find this puzzling. Later on, on line 401, it is argued that tree dominance, not only tree richness, should be included in studies. So why not here?

Lines 408-409 have already been criticised. Most of this paragraph needs rewriting. Where did the mixed forest conclusion come from – what in the results show this, if most of the forests are dominated by one species?

Line 439 states that clear-cutting is forbidden in Germany. But has it always been like this? When was clear-cutting banned in the areas studied? Can it have had any legacy effects?

Lines 456-467. Which attributes were excluded, and was it only made based on VIFs and not on biological reasoning?

Line 481. The measurement of horizontal heterogeneity is unclear to me. The plots were 1 ha in size, i.e. 10 000 square meters. So why were 25 400 m² (square meter) cells used per hectare? Then your measures are averages of partly overlapping cells. Or have I misunderstood something?

Line 563-567. The use of the Tukey's HSD tests assumes that observations are independent (I assume). So you need to adjust this somewhat to account for the non-independency of the correlation coefficients (see above). Alternatively, you could examine if each correlation (between X and Y) differs significantly after accounting for environmental differences. The hypothesis would be that they decrease significantly when the driving variable has been accounted for. You can use the test in Sokal & Rohlf (1981, pp. 583-91 according to my excel version, but I'm writing this on the train and cannot check it) outlining how to do this. I don't think this is very elegant, but it would be a way of checking the stability and consistency of the results.

Supplementary materials. Most of my comments have been discussed above.

Line S258-. Is there no information WHEN harvesting was made. It could be important?

Lind S266-. There is no information on which sources the list of edible fungi was made.

Good luck! This MS contains a very fine idea, but deserves more work.

Jan.bengtsson@slu.se

Reviewer #2:

Remarks to the Author:

GENERAL APPRECIATION

Managing forests for the supply of multiple ecosystem services is a major challenge for forest

ecologists and managers around the globe since management practices necessary leads to trade-offs amongst many services making difficult the identification of optimal practices. The authors present a comprehensive study of 150 temperate forest plots (Germany) in which the effect of different drivers (3 management types, 4 environmental factors, and 11 forest stand attributes) on 16 ecosystem service proxies were assessed. Robust analyses and overall well documented supplementary information support their manuscript. They found that forest stand attributes are the most important drivers of final ecosystem services and of synergies between services, while environmental factors are the most important drivers of intermediate ecosystem services.

This study distinguished itself from previous studies on multiservice forest management in two main aspects:

- First, the number of ecosystem services evaluated, 16, is remarkable, and is well distributed among the four categories of services (provisioning, supporting, regulating, cultural). Most studies generally compare the supply of a limited number of services such as wood volume, carbon stocking, and some measure of biodiversity, well below the amount and range of services covered by the authors.
- Second, the authors provide a novel perspective in investigating the supply of ecosystem services by looking at their relationships with environmental factors and forest stand attributes. Previous studies have focused on the effect of given management practices on services. However, the effect of management practices varies with environmental conditions and with the variety of structural and compositional stand characteristics that they generate. Here, the authors propose to focus on these intermediate drivers, with the promising hypothesis that such analysis will provide a basis to conceive novel management approaches that would target the stand characteristics associated with greatest benefits amongst different services.

Notwithstanding these advances, the study presents two important limitations that raises doubt of its suitability for publication in a high impact journal such as Nature Communications.

- First, while 150 forest plots were investigated, these plots did not cover a wide gradient of management intensities, as pointed out by the authors themselves. Additional forest plots, maybe from other European countries, to include intensive management such as clear cuts, as well as unmanaged coniferous plots, would have strengthen the study by making it more comprehensive and by potentially identifying the drivers of stronger trade-offs between services. Conclusions would have been of interest to a larger group of forest ecologists and managers confronted with strong trade-offs.
- Second, the contribution of this study is limited by the absence of a final step that would propose creative or operational management approaches based on the authors' findings. While the study reports on the most important drivers of ecosystem services, the problem of identifying appropriate management practices that limits trade-offs amongst services remains. For example, the last sentence of the introduction reads: "Our results suggest that forest management practices can be targeted to reduce trade-offs and promote synergies between ecosystem services". The authors ought to provide at least a few examples of a set of targeted management practices that together achieve a reasonable multiservice forest management approach. Therefore, when the authors state in their discussion: "our results clearly show the value of determining the effect of individual forest attributes on ecosystem service supply rather than only considering management types", I have to disagree. The authors do demonstrate the effects of forest attributes on ecosystem services, but not the value of this demonstration.

METHODOLOGICAL CONSIDERATIONS AND NEEDED CLARIFICATIONS

The importance of the temporal scale when measuring different services or evaluating the multifunctionality of forests is not well acknowledged. All services should be measured over the same time period, usually a rotation period. Since it is not the case, explanations as to why it is reasonable to do so should be provided. For example, C storage in trees is evaluated based on living tree volume while harvested timber is extrapolated from stumps, both quantities have therefore accumulated over different time periods.

The claim that forest attributes were the most important drivers of final ecosystem services while, environmental factors were the most important drivers of intermediate ecosystem services (in the abstract and result section) is too strong given that these observations are true only on average. Had the authors considered additional final and intermediate services, would these average results still hold?

In determining the drivers of ecosystem services, the potential correlation between management type, for managed forests, and forest attributes, should be acknowledged. On L288, the authors report that the only associations of harvested timber with forest attributes (deadwood volume and canopy cover) in managed forests suggest that this ecosystem service affect forest attributes rather than being affected by them. Additional negative associations with forest attributes would have appear had intensive management plots been included in the analysis.

Supplementary figure 2 indicates a non-zero harvested timber in unmanaged plots. This is non-intuitive and requires clarification.

Supplementary figure 2 indicates that pest control is much larger in managed coniferous plots than managed/unmanaged broadleaves. However, pest control has only been evaluated with regards to the bark beetles and their antagonists. Are these pests generalists or do they favour a coniferous/broadleaf tree species? It is uncertain whether this result is biased by the choice of the pest species studied.

Related with the previous comments, I wonder if the supply of certain services is de factor higher in broadleaves or coniferous forests. For example, floral aesthetic value also presents a large asymmetry between coniferous and broadleaves plots. A more solid comparison would require that managed and unmanaged broadleaves plots be compared together, and that managed coniferous plots be compared to unmanaged ones, which unfortunately, could not be done.

In the introduction, the line of arguments leading to your research objective needs to be strengthen. Emphasis should be put on one clear research objective that should be linked with one clear novel result. The introduction is long and the paragraphs are not clearly divided into the syntax rule of "one idea per paragraph", leading to redundancy. Moreover, many vague terms or statements should be clarified, eg. "different aspects", "broad types", "large range" and will help tightening the introduction.

The distinction between management type, management practice and harvesting method, should be clarified and use coherently throughout the text for the study and results to be appreciated by a broad readership. Additionally, the description of the analysed managed plots ("Managed plots in our study area include selection forestry and age class forestry, but not clear-cutting, as this practise is forbidden in Germany") should be part of the main text and not simply in the Methods section, since "managed" refers to more intensive practices (clear-cut, plantation, etc) in many countries.

SMALLER COMMENTS

L62: In the abstract, the term "forest attributes" is vague. Indications should be given to the reader as to what kind of attributes your study is referring to: compositional, structural, functional?

L62: It is important to add "stand-level" before "forest".

L63: "management types" should be defined. Again, it is vague when reading the abstract.

L68-71: The last sentence of the abstract should be improved to emphasize the novelty of the work. Presently, the described results are not very surprising and the "particular forest attributes are required for some services" is quite generic.

L96: It is missing references about studies that did make distinction between different harvesting methods within a broad management type. For example Bradford & D'Amato. Front. Ecol. Env. 2012.

L110: This paragraph would benefit from adding an example of the effect of forest structure on a recreational or cultural services, since you point out the scarcity of such studies in the 1st paragraph.

L116: These are examples of the effect of harvest methods on forest structure, and not of forest structure on services as the first sentence announces. I suggest adding examples of services to which these structures are related.

L118: Specify the kind of heterogeneity: age, diversity, vertical structure?

L123: There is an asymmetry between the examples for forest attributes and the ones for environmental conditions. I suggest adding examples for environmental conditions.

L134: This first sentence of this paragraph should be: "There is limited information on how particular forest stand attributes affect relationships between multiple services". The other part of the sentence is one of the reasons of why this is the case, and it should be in a separate sentence as done for the other reasons.

L135: "a few pairs of services" should be replaced by "a limited number of services"

L138: This sentence is vague; it needs an example.

L154: "wide gradient in management types" is in contraction with the three management types cited in the previous sentence.

L164: I find the term "region" vague, given that it is its first occurrence in the text. Are you referring to bioclimatic region or location?

L179: "forest types" should be replaced by "analysed forest plots"

L185: "effects" should be removed from "generally negative effects"

L189: "different aspects" is vague.

L190: It should be specified that forest attributes on average are good predictors of final ecosystem

services.

L204: Add "stand-level" in front of forest attributes.

L208: Reference to figure 1b should follow reference to figure 1a. Sub-figures or result description should be interchanged.

L326: "services are higher" should be replaced with "services have higher value".

L367-379: The discussion about multifunctional landscape management should include references about functional zoning, such as the TRIAD approaches. Ex: Gustafsson et al. BioScience 2012, Messier et al. Forestry Chronicle 2009.

Figure 1: Interchange sub-figures a and b. Moreover, "Management type" should be colored in red, instead of light brown.

Figure 3: The pink for "Education" is hard to distinguish from the pink of "bird-watching".

Manuscript NCOMMS-17-33179

Felipe-Lucia et al. *Multiple forest attributes underpin the provision of multiple ecosystem services.*

Response to reviewers' comments:

Reviewer #1 (Remarks to the Author):

Review of Felipe-Lucia et al.

Comments for the authors.

This is a potentially very interesting and novel paper on the drivers of forest ecosystem services, based on a large data set from three extremely well studied German sites, the so-called Biodiversity Exploratories. The data come from ≈ 150 plots in total, and are analysed in novel ways to explore the linkages of multiple ecosystem services and the environmental, management and forest structural variables that may affect the levels of and correlations between the services. It has the potential to advance the field of ecosystem services research substantially, once a number of critical things have been sorted out (see below). The authors also emphasise the management implications of their results; however, this is more contentious and will need substantial re-working and re-thinking before it can argued with confidence and communicated with forest managers and policy-makers.

Response 1. Thank you for your comments, we have now rewritten most part of the text to provide a clear message for forest managers and policy-makers (see Response R67 for more details).

I was initially impressed by this work and its attempt to shed light on ecosystem multifunctionality. The statistical analyses seem adequate, sometimes quite elegant, simple to understand and conservative, and addressing the questions well. However, I have a number of concerns, comments, questions and requests for clarifications that need to be addressed before I am able to recommend anything on this MS:

Firstly, I find the choice and classification of ecosystem services (ES) unclear and inconsistent (pt 1 below). Not only are the ecosystem services classified in a haphazard way, some of them are also questionable and cannot be interpreted as proxies for any service or ecosystem function.

R2. In the previous version of the manuscript, we classified ecosystem services in 4 categories (supporting, regulating, provisioning and cultural services) following the common framework of the Millennium Ecosystem Assessment (MEA, 2005). In addition, we split the services in two groups 'intermediate' and 'final', according to Fischer et al. (2009), a classification that has been used in many other publications because it provides an easy way to understand ecosystem services directly or indirectly used by humans. However, we acknowledge that the general reader might not be aware of this second classification, and therefore in this new version we have removed the classification into intermediate and final services and stick to the MEA classification as suggested by the reviewer in a later comment. The 'classical' MEA classification can easily be related to the more recent classification of CICES and IPBES by including the supporting services together with the regulating services (or contributions to people, according to IPBES). In general, IPBES considers MEA-CICES provisioning services as material contributions, and MEA-CICES cultural services as non-material contributions. The comparison across classification systems is now explained in the Methods section

(L. 502-506). We have also explained our choice of proxies further and have removed some proxies in response to the reviewer's comments.

Millennium Ecosystem Assessment. Ecosystems and Human Well-Being: Our Human Planet: Summary for Decision Makers. (Island Press, 2005).

Fisher, B., Turner, R. K. & Morling, P. Defining and classifying ecosystem services for decision making. Ecol. Econ. 68, 643–653 (2009).

CICES, 2016. Towards a common classification of ecosystem services. CICES V4.3. <https://cices.eu/>

Díaz, S. et al. Assessing nature's contributions to people. Science 359, 270–272 (2018). <https://doi.org/10.1126/science.aap8826>

Secondly, the broad classification of the factor “management” into 3 classes, unmanaged, managed conifers and managed deciduous forests, is too coarse to be useful and does not reflect the management practices that produce the forest attributes that are argued to be main determinants of ecosystem services. In the text, there are several examples that management actually determine forest attributes, but it is surprising that the consequence is not realised by the authors. This implies that one cannot argue that forest structures/attributes are the drivers, since they are produced by management, and can be changed by management (point 2 below).

R3. We apologise for the misunderstanding here. We fully agree that forest management drives the forest attributes and did not mean to imply otherwise. We also agree that the broad classification into 3 management classes (unmanaged, managed conifers, and managed broadleaves) is too coarse and that was exactly what we aimed to show. We wanted to demonstrate that we cannot predict forest ecosystem services simply by the forest management type and show that it is important look at the details of the forest attributes if we are to understand what drives different ecosystem services.

However, we acknowledge that this point was previously not clear and that it could be confusing for readers and forest managers. Therefore, in this new version of our manuscript, we have removed the forest management category from the analyses and have made a stronger point about the importance of analysing individual forest attributes.

Thirdly, dominant tree species is not included among the independent variables that are related to ecosystem services. It is well known that tree species are important “ecosystem engineers” that affect many ecosystem functions (which is also acknowledged in several places in the text). It may be that the dominant tree species is confounded by region in this study. If so, it must be clearly stated, but then the value of the study is in my view diminished (point 3 below).

R4. We agree that the species identity or dominance is an important variable related to ecosystem services. In the previous analyses, we included ‘proportion of conifer cover’ among the independent variables, because in our forest plots there are only 2 dominant species: spruce and beech. As beech and conifer cover are highly correlated (-0.8), we could only include conifer cover in the models. Other dominant species are Oak in 7 plots, Pines in 15 plots, plus 7 plots of mixed Pine and Beech (all of these plots are found in the North-west region) (Fischer et al. 2010).

In the new version of the manuscript, we have performed sensitivity analyses, to test if the identity of the conifer species (i.e. pine or spruce) is important, and we did not find any differences in the overall models. Although we did identify that pine cover affected pest control, soil C storage and edible fungi, and that spruce cover affected dung removal and timber production. These differences

are presented in the Supplementary Material (Sup. Table 6b) but for the purposes of clarity and brevity we retain the two species grouped as conifers, in the main text.

In addition, we also included the 'proportion of oak cover' as an independent variable in the models, which had an effect on a few services. It is not possible to fit the cover of beech in addition because beech cover is $1 - (\text{oak} + \text{conifer cover})$. We also tested species specific effects of the broadleaf species by running a model including both oak and beech cover but excluding conifer cover and did not find major differences in oak and beech effects (Sup. Table 6c).

Fischer, M. et al. Implementing large-scale and long-term functional biodiversity research: The Biodiversity Exploratories. Basic Appl. Ecol. 11, 473–485 (2010).

Fourthly, although the analyses are elegant, I am of the opinion that several of the interpretations cannot be made and that the conclusions are debatable. There are also some statistical issues that I would like to be clarified (point 4 below).

R5. We have now addressed all the points raised by the reviewer regarding the interpretation of the results and the statistical issues (See more specific responses below).

Most of the above points would have been handled, if you had thought a bit more about the system you're examining. Had you made a clear systems diagram, or thought about the questions from a structural equation point of view, you should have realised that production forests, their structure and function, are driven by management choices, and even if you didn't have full information it would have been a latent variable that drives the system. I am NOT advocating a structural equation approach, your questions are different, but you should have thought a bit more what actually produced the forest attributes you centre your MS on.

R6. We agree with this comment and we did indeed put much thought into how environmental factors, management and forest attributes should directly and indirectly drive the ecosystem services, before we started analysing the data. Broadly, our conceptual model was that environmental factors partly determine management, and that environmental factors and management jointly determine the forest attributes. We also selected forest attributes for the analysis that should be amenable to management, as it was our intention to explore whether focussing on precise forest attributes rather than management categories would give a clearer picture of how management was affecting ecosystem services. It was certainly not our intention to present management and forest features as independent drivers of ecosystem services. However, we acknowledge that our approach was not clearly stated in the previous version of the manuscript. For clarity we have therefore removed the comparison with management categories and have focussed on the forest attributes as drivers of services (L. 94-96). We hope that this makes the message and approach clearer.

I also believe that there is an unfortunate mix of attempted generality and high specificity of the results that diminish the quality of the MS. Its value lies in providing a general framework on how ecosystem services and the relations between multiple ES can be examined, given that there are a number of environmental, ecological, social, management and economic variables that can explain these relationships. But the authors also resort to discussing a number of very specific details when trying to explain the results, at the same time as they omit other details on biology and ecology. That makes the results puzzling and questionable. An example is the discussion on species composition and mixtures, which is even proposed as a management practice but not included in the analyses. I have a large number of more detailed comments (below) that, among other things, clarifies this problem. Any new version of this MS needs to have a better balance between the general and the specific than the present version.

R7. We have now completely rewritten the discussion to focus more on the general points and conclusions of our study. We have included some examples to illustrate these points, which we hope makes it easier to understand the general message. We have also addressed the issue of species composition effects, see previous responses (R4).

I have spent quite a long time on this MS, since I believe it's a very interesting one, that needs quite some more work.

R8. We highly appreciate the effort and the detailed comments provided, which we feel have substantially improved our manuscript.

1. The classification of ecosystem services is not consistent throughout the paper. They start by using the MA classification, viz. provisioning, regulating, supporting and cultural services. This classification is easily collapsed into the CICES one by combining regulating and supporting, so it's not really a problem, just a bit old fashioned as regulating and supporting services are grading into one another and often very similar. But later on, the authors use "final" ES as services that are "directly used by people", combining provisioning services such as timber production and harvesting with cultural services. This usage is awkward and conflicting with all other classifications that I know of. It is always emphasised that cultural services should be kept distinct as they are fundamentally different than biophysical ones, a notion that is also consistent with the critique on ES from social sciences.

R9. The classification into final and intermediate services followed Fisher et al. 2009. We used this classification as it is simple but makes an important distinction between services that are directly used vs. those that are indirectly used. However, given that this may not be so clear, in this new version of the manuscript we have avoided these terms and stick to the 'classical' MEA classification (see R2 above).

The problem is that I believe this greatly affects the interpretation of the results. It is argued that intermediate services (supporting and regulating) are more affected by the environmental conditions, while final (i.e. both provisioning and cultural) are more affected by forest attributes, However, from Figure 1b, it is clear that the "final services" may or may not show this pattern. C storage in trees, harvested timber show this, while temperature regulation does not, and neither does C storage in soil (because it has a large environmental component). Among the cultural services wild edible plants, bird-watching potential (i.e. number of bird species seen) and to some degree edible fungi shows the pattern, while floral aesthetic value and educational value definitely do not follow the pattern. And in Figure 1a it is clear that the two fungal diversity indicators, that are contested proxies of some ecosystem functions, do not have a large environmental component. In order to make the point they want to make, the authors have to combine two disparate classes of ecosystem services that have absolutely nothing in common, and even then the patterns are ambiguous. To propose this is a major outcome of their study, as in the abstract, is simply going too far, in particular since there is no statistics backing up this pattern. The variance partitioning does not produce any generalised statistic, as far as I could find.

In my view, it would be better to stick to the originally chosen MA classification, analyse the ES according to this, and realise that there are some services that are affected by forest attributes (which I interpret as more specific management practices, but that's another point) and some that are not. It is more enlightening to see which these former or latter services are. Or retort from advanced statistics and see if some simple things like rank tests could be illuminating.

R10. We agree that we failed to provide any statistics for the difference between intermediate and final services. In the new version of the manuscript we have followed the recommendations suggested by the reviewer. We analysed all ecosystem services together and discuss which are more

affected by forest attributes (which should be more amenable to management) and which are more driven by the environment, and therefore would be harder to alter through management.

Finally, on this point, many of the ES proxies are somewhat stretching the imagination:

To use fungal diversity as proxy for decomposition or pest suppression lacks good scientific evidence. The relations between diversity and processes cannot be assumed, it should have solid evidence behind it.

R11. We agree with the reviewer that there is some controversy about the relationships between diversity and functioning for some taxa, such as saprotrophic fungi and that diversity is too indirect a proxy of function in this case. As we have included more direct proxies of decomposition (e.g. root decomposition, dung removal), in the new version of the manuscript we have removed saprotrophic diversity as a proxy of decomposition.

Regarding pest control, we recorded the abundance of potential bark beetle pests. As bark beetles are one of the major pests in forests (Raffa et al. 2008), this should be close to a measure of pest control, as all these species are capable of causing damage on trees (see details in Sup. Methods).

Raffa, K. F. et al. Cross-scale Drivers of Natural Disturbances Prone to Anthropogenic Amplification: The Dynamics of Bark Beetle Eruptions. BioScience 58, 501–517 (2008).

Temperature regulation is questionable. Do forests really contribute to temperature regulation that has a societal value? If so, it's related to recreation, or it has a large scale effect on temperatures experienced by people in other ways. I found no references with evidence for this, and it should be discussed a bit more before accepting it as a provisioning service. Or is it cultural?

R12. We consider temperature regulation as a service because of its influence in regulating temperature and climate at the local scale. Local temperature regulation is commonly listed in ecosystem services classifications: for examples CICES acknowledges that '*Mediation of ambient atmospheric conditions (including micro- and mesoscale climates) by virtue of presence of plants improves living conditions for people*', and includes several studies that support this (e.g. Rosenzweig et al., 2006). Following other studies and reports (e.g. Easterling et al. 1997, Lee et al. 2011, Sanderson et al. 2012), we used the Daily or Diurnal Temperature Range (DTR) as a proxy of this service. For example, Lee et al. found that DTR is an important measure of surface climate variability and that it is reduced with forest cover. This index calculates the difference between the maximum and the minimum temperature each day. Larger DTR values indicate larger variation in daily temperature (and therefore less 'regulation' of the temperature), and smaller values indicate less daily variation and thus, 'more regulation'. To facilitate the interpretation on these values we use the inverse of the DTR, so larger values always indicate larger amounts of the service.

Easterling, D. R. et al. Maximum and Minimum Temperature Trends for the Globe. Science 277, 364–367 (1997). <http://science.sciencemag.org/content/277/5324/364>

Lee, X. et al. Observed increase in local cooling effect of deforestation at higher latitudes. Nature 479, 384–387 (2011). <https://www.nature.com/articles/nature10588>

Rosenzweig et al., 2006. Mitigating heat island effects in cities with urban forests.
<http://citeseerx.ist.psu.edu/viewdoc/download?doi=10.1.1.543.4848&rep=rep1&type=pdf>

Sanderson, M., Santini, M., Valentini, R. & Pope, E. Relationships between forests and weather. (EC Directorate General of the Environment, 2012).
http://ec.europa.eu/environment/forests/pdf/EU_Forests_annex1.pdf

Harvested timber is somewhat problematic. It does not measure the supply, but rather of demand – the potential supply will be there even if the demand disappears. The argument that timber supply in unmanaged forest is zero is wrong. It's the demand that doesn't exist. In my view, biomass production would be a better proxy for potential timber harvest, i.e. supply, but if this supply is used is another thing. But this is contentious, however. I think you may need a better argument. But in addition, the variable is calculated from the stumps found in the plots. In order to know what this variable means (and hence the time frame of this variable) one needs an estimate of the decomposition rate of tree stumps. Is it 20 or 50 years? This makes a large difference for understanding this proxy. If stumps don't decompose for 50 years, it is a decent proxy of historical harvesting rates, but if stumps disappear in 20 years time, then it's not producing an indicator of the harvesting rate that produced the structures/attributes analysed.

R13. We do agree that a measure of potential supply is more appropriate, given that all of our other proxies are of ecosystem service supply. In this new version of the manuscript, we have updated this variable and used the mean annual increments (MAI) of wood volume, averaged across the rotation period for even-aged forests and the periodic annual increment (PAI) between two forest inventories for uneven-aged and unmanaged forests. We believe this new measure is more appropriate because it quantifies timber supply independently of its demand, and therefore, is more comparable to the other proxies.

Edible fungi has a list of "edible species", although there is no source for the information. However, many of the "edible species" are of the Amanita genus, which contains many poisonous species and it is generally argued that they should not be picked at all because of the risk of confusing them with highly poisonous ones. One species is redlisted (battarae) and two are likely somewhat poisonous (fulva and rubescens). The other species in the list look acceptable, however. But, how many of them are often and likely to be picked by the public, and is it a provisioning or recreational service?

R14. We agree with the reviewer that experienced knowledge is required to collect some edible mushroom species. However, in this manuscript we are only assessing the potential of finding edible mushroom species in the forest, and we don't assess the likelihood or amounts of mushrooms collected (this would measure the demand for the service, while, as noted above, we focus on potential supply here). We consider this proxy as a 'cultural' service because in Germany people largely collect mushrooms as a recreational outdoor activity, rather than as a major source of food.

Regarding the classification of edible fungi, the previous version of the edible fungi species list was based on the expert knowledge of a professional Mycologist and member of the "Swiss Union of official Mushroom controllers" (VAPKO; <http://www.vapko.ch>). In this updated version we strictly followed the criteria of the German Mycological Society (DGfM), excluding those species with inconsistent edible value (see references below). Therefore, the updated list of edible fungi has been drastically reduced (See updated Sup. Methods for details) and several of the species the reviewer mentions have been omitted.

*Deutsche Gesellschaft für Mykologie e.V. (DGfM). 2015 'Positivlist der Speisepilze'.
<https://www.dgfm-ev.de/speise-und-giftpilze/Speisepilze>*

(link to online version of the pdf: <https://www.dgfm-ev.de/speise-und-giftpilze/speisepilze?reattachment=521d327db648f678d7def3698dd76a4a>)

(link to online pdf of sp with inconsistent edible value: <https://www.dgfm-ev.de/speise-und-giftpilze/speisepilze?reattachment=59540261ada01613b7cd4c54784db6c3>)

The plant lists are quite overlapping, and I am a bit surprised that these three services are treated as separate as they partly rely on the same species, but I accept that – although it means that they are not independently estimated. More worrisome is the fact that “floral aesthetic value” is only comprising the spring-flowering species and not, for example, the 6 orchid species in Table S13, nor *Fragaria*, *Hypericum* or *Prunus* from Table S11. It seems to me that not much thought has gone into these proxies, but perhaps the “experts” used differ from me in this respect. But really, these cultural services should have been evaluated together with stakeholders, like amateur botanists, people using forests for recreation, etc.

It is good to try to examine many ES, but the question is how much evidence you need to do it. In my view, some more caution is needed for some of the proxies.

R15. Thanks for making this point. We agree that there was some overlap in the species list, therefore, we have now replaced the indicators ‘floral aesthetic value’ and ‘educational value’ by a new indicator for plant species with cultural value. This list considers all plants of interest either because they contribute to the forest's aesthetic value (e.g. species blooming in spring), or because they are of interest to botanists (e.g. because of their beauty or rarity). Species of special interest for the general public or for botanists were identified by botanists from the Botanical Society of Bern (Bernische Botanische Gesellschaft), an amateur botanical society whose members are also knowledgeable about preferences of the general public. We do fully agree that an in-depth consideration of stakeholder preferences would be desirable but this would require stakeholder interviews and is outside the scope of this work. We hope that following the consultation with an amateur botanical society we have a list that includes those plants of particular botanical interest to the public.

2. The broad classification of the factor “management” into 3 classes, unmanaged, managed conifers and managed deciduous forests, is problematic. It is too broad to be useful, especially since you actually want to make a case about which ES depend on management (well, here you mainly state ES do not depend on management). Of all the ES, only harvested timber is largely explained by management (Fig 1) but that's because unmanaged forest cannot produce any timber (according to the definition of the ES). But these three categories do not adequately reflect the other management practices in these forests. It is management that produces the forest attributes that are argued to be main determinants of ecosystem services. In the text, there are several examples of this:

Carbon storage in trees, for example, is of course directly related to forest management, in the sense that management is determining how much is harvested (not only if, as the three categories suggest) and when (i.e. DBH) the forest is harvested. All ES that depend on forest structure and canopy cover, e.g. temperature regulation, the plant ES, are determined by management, not by the structures that management has produced – i.e. it is harvesting, thinning, and other management interventions that are important, THE ATTRIBUTES ARE ONLY PROXIES OF MANAGEMENT INTERVENTIONS. Also, they come in clusters related to management practices, but it is not clear what these clusters might be since the full data are not included.

R16. We fully agree that management is one of the key drivers of the forest attributes, as discussed above (R3). However, not all forest attributes are strongly affected by the management type; as we can see in Supplementary Fig. 1, some forest attributes have similar values in different forest management types (e.g. high canopy cover values in both managed conifer stands and unmanaged beech forests), this emphasises the importance of looking at the individual forest attributes to understand service provision. We fully agree that the three management categories are too broad to be useful and this was the point we wanted to make but it is apparent this did not come across clearly enough and we understand that is a major concern of the reviewers. Therefore, in this new

version of the manuscript, we emphasize the relationships between forest attributes and ecosystem services and do not discuss the management categories.

Additionally, we have included a new Supplementary Fig. 3, where we show the correlations among the 21 forest attributes and the 4 environmental variables.

So, management actually determines forest attributes, and it is surprising that this is not realised by the authors. One cannot argue that forest structures/attributes are the drivers, since they are produced by management, and can be changed by management, as indicated in the text later on, on e.g. lines 286-289, line 305-312, lines 350-352 where it is argued that open canopies and disturbances are due to harvesting, lines 396-399, etc. On line 401, it is not realised that tree richness (and tree species composition) is also determined by management in most forests (during thinning, for example). And on line 417 mixed forests are suggested to promote some services – but mixing species is also a management practice not caught by your “management” factor.

It could be that German forests are managed in such a simplified scheme as suggested, but if so – which I doubt – then this study is just a case study of a particular way of managing forests that has limited general relevance. I hope this is not the case, because I believe the German model as well as the Biodiversity Exploratories have a lot to contribute in general.

I needed to explore this in some detail, because it means that your conclusion that forest attributes is more important than “broad forest management types” (line 409) confuses the important issue and makes this study of quite small policy relevance. It confuses (well, hopefully not) policymakers into stating that management is unimportant for forest ES, when in fact it is, and any forest manager knows this. You really need to address this in a more clever way, and start from the obvious fact that management in production forests is a main determinant of forest structure and functioning. The unmanaged forests here are interesting, but irrelevant for production forests, which still is the main proportion of European forests, and something that we can influence.

R17. We fully agree, we certainly did not mean to suggest that management was unimportant for forest ES, just that management categories (as used in many studies) are not helpful proxies of ES supply. By focussing on the forest attributes and removing the categories (see above), we hope that the message is now clear: forest management is a key driver of ES and their relationships in forests and management of particular forest attributes could optimise the supply of multiple ES. We also discuss the effects of management on forest attributes in the methods (L. 565-575) to make the point that attributes respond to management more clearly.

3. A major drawback, that limits the applicability of the study, is that dominant tree species and species composition were not included among the independent variables that are related to ecosystem services. You only use the less ecologically informative tree species richness. It is well known that tree species are important “ecosystem engineers” that affect many ecosystem functions. There are many studies of the effects of single species like beech, spruce, pine and other species differ in their effects on various ES (see e.g. experimental studies carried out in the 1960-90ies). This fact is also acknowledged in the text.

I could imagine that dominant tree species is confounded by region in this study, but that’s no excuse to omit the probably most important (and managed) variable that produces variation in ES between plots. It could also have been excluded when environmental variables were selected – there is not any information on which variables were discarded and why – but using mathematical/statistical algorithms to exclude the most biologically relevant variables seems to me a bit thoughtless, to say the least, so I hope you didn’t do this.

I must remind the authors that in most biodiversity-ecosystem functioning studies, species composition is more important for ecosystem functions than species richness. So there are strong reasons to use species composition or dominant tree species in your analyses. Since I would guess that there are at least three species – beech, spruce(s) and scots pine (perhaps other pines) – that are dominant, the use of species identity links directly to traits of species and thus to important ecological mechanisms underlying the effects on ES.

I am of the opinion that the analyses need to be redone with dominant tree species and/or species composition as explanatory variables. If not, the authors have to have VERY GOOD arguments to not do it. And not doing it diminishes the value of the study greatly.

R18. We fully agree that tree species composition is a key driver of forest ES. As mentioned above (R4), we previously fitted conifer cover as a measure of composition but to explore the effects of individual species even further, we now also test the effects of spruce and pine separately, together with the effect of oak cover, as well as the effects of beech and oak separately from conifers. This provides some interesting further results but does not change the main picture dramatically. Given the small number of dominant tree species, it seems that the major compositional difference between the stands arises from whether they are dominated by beech or by conifers (the beech effect is therefore simply the inverse of the conifer effect). We explain now that accounting for the cover of both oak and conifers is representative of the cover of the main tree species of our study area and avoids collinearity in the models (which would happen if beech cover was included).

4. The points 1-3 above lead me to the conclusion that several of the conclusions are not warranted, and not really supported by the data. It is not correct to state that forest attributes are the main drivers of intermediate ES, because the attributes are largely proxies of past management. The amalgamation of provisioning and cultural services into “direct” services is confusing and inconsistent with the MA classification used and much of the other work on ES. The statistical support for the two conclusions is not convincing, because no real statistical analysis was made of these patterns.

R19. We feel we have addressed all these points in the new version of the manuscript (see above) and hope that the new approach will satisfy the reviewer.

The most interesting part of the study is the attempt to find out which variables were the main determinants of ES by using analyses of the residuals from models including and excluding environment, “management” and (only including) attributes. I thought this was an elegant idea, but still there are a number of statistical issues that I need to bring up.

Firstly, since you are dealing with a correlation structure, the correlations in that structure are to some degree dependent on each other. For example, if A and B are positively correlated, then if C is negatively correlated to A, it is less likely that it’s positively correlated to B. So when you test the significance of correlations in Figure 2, these correlations are not independent of each other, and you cannot (without some adjustment or randomisation) use the Tukey HSD test.

Moreover, with respect to Fig. 2, the conclusion that “forest attributes influences the negative correlations between immediate services (implicit: more than environment)” is not warranted. Neither is the statement “attributes were ... main drivers of synergies between final services”. While I agree that the pattern for synergies in Fig. 2a suggests that environment is the most important, neither Fig. 2a trade-offs or Fig. 2b synergies show anything else than a gradual decline as more variables are added, they DO NOT show any difference between E+M+F and E+M or E (b is NOT different from a, b). The results do NOT show that the addition of F in E+M+F is different from E alone, so you can’t make any inference that the addition of F is more important than E alone, especially in the light of the Tukey tests likely being overestimating differences. This is a VERY elegant

way of testing the hypothesis, but it does not support the conclusions, apart from clearly showing that addition of E decreases correlations between synergies.

R20. In the new version of the manuscript we have removed the factor 'M' (management) from the analyses and the classification into intermediate and final services, so the interpretation of the results is much more straightforward.

In addition, we have now used a test to compare correlation matrices based on the R package *lavaan* that accounts for the lack of independence among pairwise correlations within a matrix. We estimated four different indices to assess significant differences between correlation matrices after filtering out the effects of environment or forest attributes (as is done in structural equation modelling to compare observed and predicted covariance matrices). The updated results show again that the environment (E) and the forest attributes (F) explain the synergies and trade-offs between ecosystem services, but we were not able to identify significant overall differences between the effects of environment and forest features, probably because of the high shared variances between F and E. However, we also did a complementary analysis in which we dropped one category of ecosystem services at a time from the full set and with this analysis we found that when the supporting ES were removed, there were significant differences between E and F, suggesting that supporting ES are mainly affected by the environment, while regulating and cultural services respond also to the forest attributes. We confirmed this finding by randomly removing sets of 5 services (equal to the number of our supporting services in our study), 4 (equal to the number of regulating and cultural services), and 1 (equal to the number of our provisioning services, timber production), and found that randomly dropping ES did not have any effect on the differences, i.e., there were no significant differences between E and F. This new analysis is presented in L. 238-249 (and Sup. Table 4 a,b) and confirms that the effect we found was linked to the supporting services. We also look at significant changes in individual correlations, to further understand which trade-offs and synergies are driven by the attributes or the environment. We use the package *cocor* which provides tests of the significance of differences for individual correlations and corrects for the non-independence (L. 253-260, Sup. Table 4c).

If I am right, this questions the analyses of forest attributes on final ES reported on line 271 onwards, because here the authors assume that final ES are driven by forest attributes, but as I have argued this is not warranted from neither Fig. 2b nor Fig 1b. While this analysis is interesting, the two Figures rather suggest that environmental variables should be included in a more full analysis of what drives ES.

R21. We have now included all ES in Fig.3, i.e., we analyse the effect of each forest attribute on all ES. The point of Fig. 3, and of our analysis in general, is to determine how forest management affects ES through changing forest attributes. We would therefore prefer to focus on the effects of the attributes and not of the individual environmental factors, as we feel that showing the effects of these factors as well would make the results too complex. We also feel that the environmental effects are less interesting in the context of our questions because they cannot readily be affected by management.

We use variance partitioning analysis to identify the services that are more affected by the environment and so are less amenable to changes in management. We have now included a table with the AICc and the estimated effects of each predictor in each model (E, F, E+F) in the supplementary material (Sup. Tables 3 c-f).

Furthermore, in Table S5, the authors adjust the P-values for multiple comparisons, but this is not the problem. The authors should have adjusted the P-values for the non-independence of the correlation coefficients before doing anything else. They need to figure out how.

R22. We agree and as explained above (R20), we have now used a test to compare correlation matrices based on the R package *lavaan* and have estimated four different indices to assess significant differences between them. In addition, we also tested for significant differences between correlation matrices for each pair of ecosystem services using the package *cocor*, which corrects for non-independent observations.

In Fig. 2, the standardised effects are shown, which is OK given the analysis, but it does not show any information on the actual relations in the field, which from a more practical point of view would have been desired. But this is just a small issue.

R23. We show standardised effects, so that it is possible to compare between ecosystem services, in terms of how they respond to the same forest attribute and to compare between attributes in terms of the services they affect. It would not have been possible to determine which features were the strongest drivers of a particular service proxy, or which services were most strongly affected by a given feature, if we had shown the raw, unstandardized effects due to the different scales of measurement of the different services and forest attributes. However, we do agree that it is interesting to show effects also on the original scale and so we have now included a figure with the raw effects as Sup. Fig. 5. This is a much more complex figure because each service x attribute relationship has to be plotted separately, it is of course not possible to plot multiple effects for a given service on the same graph, due to the different scales, and we would therefore prefer to keep the figure with standardised effects in the main text.

Similar to this, the use of z-scores is good for standardising the independent variables, but since it makes them equal in the analyses it does not include the fact that they may have very different degrees of variation (as suggested in Fig. S1, where some box plots shows large and some small variation). This means that the major long gradients are downweighted relative to shorter gradients. Does this matter? Perhaps. But for a good understanding and hence providing management and policy advice, it is important to understand this in more detail. You need to consider the possibility that your analyses may overestimate the importance of short environmental gradients.

R24. As the reviewer points out, z-scores are used to be able to compare variables that have very different ranges, as in our case. The magnitude of effects of different attributes cannot be compared if variables are on the original scale. It is certainly true that the effects of variables which have a smaller range will seem larger if all variables are standardised but it is not possible to address this unless we standardised the attributes by the theoretical range of each variable, which would be hard to calculate for some of the attributes in particular.

However, as previously mentioned, we agree that the unstandardized effects are interesting, if the reader wants to compare to other studies, and we have now included a figure in the supplementary material showing these (Sup. Fig. 4). The unscaled ranges for the forest attributes and ecosystem services are also shown in Sup. Fig.1 and 2, which also allow the reader to judge whether the ranges in some variables are short compared to the overall range expected for each variable.

In the variance partitioning, you are using 11 attributes but only 3 management and 4 environmental factors. I am not a real expert on this, but I have understood that it is important to have similar numbers of variables in the different larger categories when doing the variance partitioning, otherwise you give undue weight to the category with the largest number of variables. You state that you solve this by using adjusted R-squares, but I'd really like you to consult a statistician on this before you make this statement. Or perhaps you already have done, in which case it should be stated in the supplementary materials.

R25. The use of adjusted R squares is recommended when the number of variables in each group is different (Peres-Neto et al. 2006), as adjusted R squares, like AIC values, “punish” the models with more predictors. We have now included a reference to support this in the methods section.

Peres-Neto Pedro R., Legendre Pierre, Dray Stéphane, and Borcard Daniel. 2006. Variation partitioning of species data matrices: estimation and comparison of fractions. Ecology 87: 2614–25.

Furthermore – and I guess this is a statistical point to some degree - the most thought provoking and interesting result (despite my criticism of it) is that you can (perhaps, and my interpretation) take the full set of correlations between ES and sequentially decompose it into those determined by the naturally given “environmental” factors, which cannot really be managed on the short time scale, those determined by “management” as such, and those dependent on those structures/attributes that are (my interpretation) created intentionally or unintentionally by management practices. This idea is very clever, but I would not have hidden it in the supplementary material. It is actually the main novelty of the paper (in my view) and deserves more than the ambiguous Figure 2. Can it be shortened and placed in the main text as a Figure or Table? And what do we learn from it? Which ES are synergistic because they are driven by the same environmental drivers, and which are synergistic after these have been taken away, and thus can be influenced by management (since after E+M+F very few are left), which is what is needed to influence policy. A table or the set of correlation matrices would be a better presentation than the tedious discussion in the text, lacking good visual images or a table, as it is now. I suggest you consider this.

R26. Thank you very much for appreciating the novelty of the method proposed and for suggesting a better way to represent it visually. We have now followed your suggestion and included the figure that was in the supplementary material in the main text. We have done so in a multi-panel figure including all correlation analysis together with the original Fig. 2 (new Fig 2). These results clearly show the importance of both the environmental variables and the forest attributes in driving the synergies and trade-offs between ecosystem services, and the particular influence of the forest attributes on the regulating and cultural services. As mentioned above, we also test which correlations are significantly affected by the environment or the forest features (which we agree largely represent the effect of management) and show this in the figure. In addition, we have now included a paragraph summarizing which synergies and trade-offs are mostly influenced by the environment and which ones can be modified through management (L.253-260). We have also calculated the change in correlation strength due to environmental factors or to forest attributes, and presented it on Fig.2c,d to facilitate the interpretation of the results. We hope that this now better displays the effects of this analysis.

All this boils down to me being very uneasy about the conclusion on lines 408-409 that forest attributes are better predictors of ES than “broad forest management types are”. Although one could argue that this is formally correct given the included variables, analyses, and their interpretation (which I am sceptical about, see above), it gives the impression that management is not very important, which is wrong. As argued above most attributes are produced by management, and the statistical support for the conclusion is not very convincing when being scrutinised. Moreover, it should be pointed out although elegant, the approach in this MS is correlational, and that experiments would be needed to address some of the major propositions/hypotheses that emerge from the analyses.

R26. We agree that management is very important, and that the effects of the forest features show the effects of management, and we hope that the new presentation of the results makes this clear. In addition, we fully agree that our study is correlational and we therefore acknowledge that further

experimental studies and studies in other forest types should be conducted to confirm our hypothesis in L. 447-452.

5. Before going into minor details, I want to point out some other problems:

The idea of “intrinsic” trade-offs is confusing and ambiguous. I assume you mean that these are trade-offs that result from basic ecological or biophysical limitations or processes, such as the examples on litter decomposition and C accumulation (line 143) or on mycorrhizae and nitrogen availability (line 251). I think, however, that the term intrinsic is confusing, when it rather should be called something like “unavoidable trade-offs for physiological or ecological reasons”.

R27. The term ‘intrinsic’ or ‘inherent’ trade-offs has been used in the literature before in other contexts, for example to highlight the difficulties in managing for biodiversity conservation, economic return and social equity (e.g. Halpern et al. 2013 PNAS). In the ecosystem services literature, the idea of ‘intrinsic’ trade-offs is used quite widely and is present in several publications (e.g. internal variables or dynamics, Bennett et al. 2009; direct drivers, Rocha et al. 2015). We feel that we need a concise term for the idea that some trade-offs arise from direct interactions and are not caused by shared responses to environmental or management drivers. We would therefore prefer to continue to use the term ‘intrinsic’, but we have now provided a definition to explain its meaning, as the reviewer suggested (L. 127-130).

Halpern, B. S. et al. Achieving the triple bottom line in the face of inherent trade-offs among social equity, economic return, and conservation. PNAS 110, 6229–6234 (2013).

Bennett, E. M., Peterson, G. D. & Gordon, L. J. Understanding relationships among multiple ecosystem services. Ecol. Lett. 12, 1394–1404 (2009).

Rocha J, Yletyinen J, Biggs R, et al. 2015. Marine regime shifts: drivers and impacts on ecosystems services. Phil Trans R Soc B 370: 20130273.

The environmental variables are presented in the main methods section, but the descriptions of the ES proxies are relegated to the supplementary materials. This is inconsistent, since the paper is about ecosystem services – not environmental variables.

R28. We agree that the methods description of ES should be in the main text. The reason that we put it in the SM was that we need to cite many references for the various different methods. As Nature Communications has a restriction on the number of references allowed, we had to move the ES methods to the supplementary material. If the editor agrees we would be happy to include the complete methods on ES in the main text, although this would increase the number of references quite substantially (40 additional references).

Furthermore, it is unclear which of the original forest attributes were not included in the final analyses, and how they were related to the ones selected. You should, in the supplementary materials, present a list of the original 20 and a correlation matrix that indicates how they related to the ones you finally used. And, as indicated above, I suspect that some of the excluded variables would be more biologically/ecologically relevant – but I may be wrong. You also need to explain better why the diversity measures used are adequate proxies of ecosystem services (see above).

R29. We have now included a Sup. Fig. 3 with the correlations of the 20 variables initially explored + oak cover that was included in this updated version of the manuscript. As we explained above, the 20 initial variables were selected to provide unique information about the composition of a forest stand. Then, to avoid collinearity among the variables, we used variance inflation factors to exclude variables that are highly correlated with the others. The final selection was made to make sure all the

variables are ecologically relevant, have a link to management and have a clear interpretation. The excluded variables were mostly derived parameters from the ones included or are very closely related to the ones included.

6. Minor details in the text:

Line 84. The term “optimise” is ambiguous. In the case here, the production of marketable timber is rather maximised and for profit but under the constraint of national laws and regulations, so any optimisation is with respect to profit maximisation.

R30. We agree with the reviewer that the term “optimise” may be ambiguous, however, amongst the coauthors we have extensively discussed what the appropriate term should be and in this particular case we consider that “optimise” fits better than “maximize”. Maximization would mean that the forests had been managed to get the highest possible yield of a single service, here timber production. However, this was not the case (and is even less so today). In Germany, in the past 50 years there have always been at least a few ecosystem services simultaneously considered by forest management. Therefore, we believe that “optimize” is a more appropriate term as it does not imply that profit maximisation is the only goal of forest management.

Line 88-90. The examples of trade-offs between intensive timber production and other services are not the best ones. What is it that is sacrificed when intensive production is performed? It is the supporting and regulating (and cultural) services, according to UKNEA and other reports. But you only use examples like water availability (apparently ambiguous), biodiversity loss and cultural services.

R31. We have now included more examples to show that timber production trades off with many regulating (e.g. carbon sequestration, water availability) and cultural services and with biodiversity conservation (L. 87-89, L. 96-105).

Lines 94-102 (approx). The question here is if the variation this study is based on is large enough to address these issues. Many boxes in Fig S1 are fairly narrow which suggests gradients are not long, with the exception of Conifer cover which instead lacks the middle range and seems to be either 80-100% or 0-10% (mainly).

R32. Although we agree that our study does not include the complete range of variation in all the 12 forest attributes, it does contain a range in these attributes typical for central European forests. Without an experimental study, which separately manipulated the various features, it is challenging to produce the complete range in all features. We acknowledge in the text (Discussion, L. 435-447) that a larger gradient might have shown stronger effects of some features. However, we believe our study is valuable because it includes a larger number of ecosystem service proxies and tests their response to a wider range of forest attributes than previous studies.

Line 105. Again the term “optimise” emerges from nowhere. What is supposed to be optimised here, and for whom? Landowners, forest industry, the public, or society? Since there are multiple interests in society concerning how forests should be managed, optimisation is always with respect to something. And ecosystem services are often regarded as public goods, so the “optimum” depends on how power over management is distributed between landowners and society.

R33. We agree that the point was not clear enough. We used the term ‘optimise’ in a general way, meaning not maximizing a particular service but maximizing the number of ecosystem services. We have now rewritten this section to clarify the point (L. 96-98).

Line 115-123. Here it is clear that management determines structures and attributes, but then you forget this in the rest of the MS. These examples of management are not caught by your 3-level “management” variable. (this is also relevant for lines 152-160).

R34. We agree with the reviewer and we hope that this point is now clear throughout the manuscript.

Line 161. What does an evenness measure mean, in forests where dominance structure is clearly very uneven (text and e.g. Fig S2 conifer cover panel).

R35. We agree with the reviewer that most (but not all) forest plots are uneven. The point of including evenness and richness was to account for these two main aspects of diversity separately in order to understand their particular effects on ecosystem services. As the reviewer correctly points out, the range in tree species richness is very low, so evenness gives additional useful information and mixed spruce/beech plots (for instance) can have high evenness. Note that we calculated Pielou's evenness which ranges between 0 and 1 and is uncorrelated with species richness.

Line 169. For whom and on which scale is temperature regulation interesting? Does it have local, landscape or even regional impacts?

R36. Many studies and reports highlight the importance of temperature regulation as a key forest ecosystem service acting at the local and landscape scale but with consequences at the global level (see R12 above).

Line 174. I have already discussed that it's not useful to combine provisioning and cultural ES. Remember this!

R37. We have followed the suggestion provided by the reviewer and have removed the conflicting classification.

Line 194-196. The recommendation on forest management practices suggested here is without substance when the results are scrutinised (also relevant for lines 201-202). It is not clear which management practices you mean, and you are in other places rather arguing that management is not important, so the paper is inconsistent on this issue.

R38. We hope that we have clarified the message in the new version of the manuscript and avoided misunderstandings with the new approach.

Lines 206-210. The amount of variation explained by the models varied quite much – from 20 to 80% for individual ES. It is evident in Figure 1, but should also be pointed out here.

R39. We agree with the reviewer that this is an interesting result that should be pointed out in the text. We have now included this information in line L. 195.

Further, are there any statistics supporting that forest attributes or environmental factors really explained MORE than the other variables in the two cases? If not, the conclusions should be weaker.

R40. We did not perform a significance test, but variance partitioning already indicates which variable explains more variance (see Sup. Table 3a,b). In addition, we have now included several supplementary tables showing all effects found in the models and the AIC of the models (Sup. Table 3 c-f). (See R21 above).

Figure 1. This figure shows the results to be much more variable within your categories than you argue in the text (lines 206-216). So there is a need for a statistical evaluation of the results from the

variance partitioning. Note that I suspect (I may be wrong) that there is really no statistical result that backs up the statements you make.

R41. This comparison has now been removed from the paper. All ecosystem services are now presented in the same figure, so no further statistics are needed.

Figure 2. The results are not showing what you state they do, since the Tukey tests are misinterpreted, because the correlations are not independent data points, and because there is mostly no significant differences between correlations when just E is included and when E+M or E+M+F has been accounted for..

R42. We thank the reviewer for pointing out the problems with the Tukey test and we have now used a test designed to compare correlation matrices, from the package *lavaan*, which accounts for the non-independence of the correlation coefficients (See R20 above). Our results were robust to these additional analyses and have not changed our conclusions.

Lines 250-254. The “consistent” correlations are sometimes expected, sometimes seemingly haphazard (like saprotrophic diversity vs. dung removal, mycorrhizal diversity vs pest control), and sometime because the ES proxies are using partly the same underlying data (the plant-based ES). But what confidence should we have in these patterns? Are there non-measured variables that could explain them?

R43. We agree with the reviewer that it is possible that other variables not measured in our study could contribute to explain these correlations. However, we are quite confident that we have included the most relevant variables for most ecosystem services, as our models explained more than 45% in variation them (except for mycorrhizal diversity, where we could only explain 21%). We acknowledge in the text that other unmeasured variables could contribute to driving synergies and trade-offs, particularly for services where there was a larger fraction of unexplained variation (L. 357-361).

However, the fact that several correlations remain after removing the effect of the environment and forest attributes suggests that there may be some ‘intrinsic’ trade-offs that were not previously identified. Our main aim here is to highlight such relationships, which could be tested further in future studies and experiments, and to provide new tools to explore the potential existence of such trade-offs. This is now made clear in the text (L. 366-375).

Line 280. Are there any possible mechanisms for tree evenness having negative (or positive) effects on ecosystem services? Or what is going on here? Is it a useful variable given that the forests are dominated by mainly one species?

R44. In our forest plots, high tree evenness occurs when there is an even mix of two species, usually a conifer and a broadleaf species. In theory such mixing of tree species could provide opportunities for complementarity between the species and could therefore increase several ecosystem services. Evenness might be important alongside species richness, as there might be few opportunities for complementarity in a species rich but highly uneven community and some biodiversity experiments have shown that evenness is an important driver of functioning (e.g. Wilsey and Potvin, 2000). On the other hand, negative effects are possible if services are mostly promoted in pure stands (e.g. van der Plas et al. 2016). Our results indicate that evenness does not have a significant on most ecosystem services, and was only found to decrease edible plants and tree C. The first case can be due to the fact that edible plant species grow better in conifer stands than in mixed forests (and in fact conifer cover has a positive effect on this service). In the second case, tree C was also found to

decrease with conifer cover and increase with canopy cover, which explains why mixed forests of conifer and beech, which have lower values of canopy cover, have reduced tree C.

Wilsey, Brian J.; Potvin, C. (2000): Biodiversity and ecosystem functioning: Importance of species evenness in an old field. Ecology 81 (4), pp. 887–892.

van der Plas, Fons; Manning, Peter; Allan, Eric; Scherer-Lorenzen, Michael; Verheyen, Kris; Wirth, Christian et al. (2016): Jack-of-all-trades effects drive biodiversity-ecosystem multifunctionality relationships in European forests. Nat Commun 7.

Lines 281-288. The amount of detail here is bewildering and confusing, although many of the relationships look like they are real and not artifacts, and driven by e.g canopy cover and light competition. However, I can't help pointing out that again the authors discuss the effects of harvesting activities, i.e. management, affecting ES.

R45. We have now removed these lengthy explanations of the results.

Figure 3. The panels are too small, and it's not reader-friendly to use exactly the same type of lines – they should be distinguished both by line type (whole, broken, dotted) and colour. What is the 95% CLs referring to? Slope? If so, should it not be larger at both ends.

R46. We have now increased the size of the panels and used a different type of line to better distinguish the ecosystem services. The 95% confidence intervals are intervals around the predicted values, calculated with the *visreg* package.

Line 305-308. The effect of DBH suggests that age of the forest has an effect, but it's not included as a variable. Why?

R47. DBH is strongly correlated with age, meaning that we could only include either age or DBH in the model. Age was discarded in the very preliminary analyses because it is not a direct measure but derived from DBH and height, and thus it was not included in the 20 initial variables considered for analyses (see new Sup. Fig. 3). We discuss in the text that many of the effects of DBH may be related to tree age (e.g. when we refer to old-growth forests). In addition, we explain now in the methods that DBH is often correlated with age, particularly in our case study.

Line 310-312. Here the authors bring up managing for mixed forests, but there is nothing in the results that suggests anything about mixed species forests. And is the introduction of small-scale disturbances feasible without any ideas about which actual management practices are producing which structures?

R48. We acknowledge the term mixed forest was not clearly explained and have now rephrased the discussion to refer to forests that are mixed in terms of their attributes, not necessarily in terms of their tree species (L. 389-391). We now make this point more clearly and emphasize that our results show that different forest structures increase different ecosystem services, and therefore, that a mixture of forest types, resulting in a high diversity of forest structures at the landscape scale, should increase the range of ecosystem services provided.

The forest attributes shown to have positive or negative effect on ecosystem services can be achieved using different forestry techniques, and it is not the aim of this manuscript to discuss which techniques are more appropriate to obtain the desired forest attributes, however, we feel that it would be feasible to promote small scale disturbance with targeted removal of trees for instance. We also include more discussion on the relationship between management and the forest attributes to make it clearer how they related to management practices (L. 391-396, and also L. 565-576).

Line 324. The synergy between carbon storage and temperature regulation is most likely due to both being related to a third variable. C storage and temperature regulation are acting on entirely different time-scales, so it's highly unlikely there is a direct relationship between them. The closed canopy explanation is not on the time scale of C storage – rather C storage reflects the history of land use during the last 50-100 years.

R49. In this case, we are not accounting for the variability of temperature along the rotation period or life span of a forest plot, but on the relationship among these two variables in the present time. If we go now to a forest stand and estimate the wood volume and the C stored there, we see that this correlates well with the temperature buffer effect, simply because larger trees store more C and have a large canopy cover, which provides this insulation effect (Easterling et al. 1997, Lee et al. 2011, see R12 for references).

Line 329-30. The two explanations put forward are both possible, but very different biologically and management-wise. If the result still remains after new analyses you may have to think about this.

R50. The relationships are consistent after re-analysing our data and we have thought about this further. We are quite confident of the explanations provided, which are supported by a number of studies cited in the text.

In general, the discussion is too long and has an imbalance between detail and generality that I mentioned earlier.

R51. We have now shortened the excess of details in the discussion and presented the key messages more clearly.

Line 355. The cultural services are not driven by the species of interest, but rather by the interests of particular people or stakeholders, for instance those interested in recreation or bird-watching. And there's not societal interest in these, rather just a stakeholder interest (which is good enough to call them ES, though).

R52. We agree with the reviewer and have now rewritten this section to clarify the message (L. 382-387).

Line 357. *Vaccinium myrtillus* and *Rubus idaeus* are two plants with entirely different biology and traits, and unlikely to be affected by the same management. *V. myrtillus* is more of an old-growth plant, and *R. idaeus* a clear-cut or disturbance-favoured plant, at least at more northern latitudes.

R53. We have now cut this more detailed discussion of the edible plants, in order to focus the discussion on the more general conclusions of our study.

Line 358. The “appealing plants” are all to be considered “spring-flowering” which is a bit awkward since many other plants among those in your three lists are to be considered appealing, e.g. the orchids.

R54. We agree with the reviewer and have now replaced the indicators ‘floral aesthetic value’ and ‘educational value’ by a new indicator of plant species with cultural value, which considers both plants of special interest for a wider range of people, from recreationists (e.g. species blooming in spring, to botanists (e.g. because of their beauty or rarity). We have provided a full response to this point above (R15).

Lines 371-376. This part needs to be re-thought after new analyses have been made. The ideas on landscape management are fine but may need re-thinking.

R55. We have re-done the analyses and the results are robust to the changes made. We have reflected on this and are confident that the ideas discussed in this section are valid and accurate.

Lines 384-387. These are key lines in my reasoning, but they are unclear on what they mean. It does argue that these forest plots are indeed dominated by one species like beech, and with only few individuals of other species. Hence tree species composition, least if including relative abundances, is likely to be very important, but still it is not included. This is in the light of these lines a bit surprising. And despite the problem written out with the tree diversity variable, it is retained. I find this puzzling. Later on, on line 401, it is argued that tree dominance, not only tree richness, should be included in studies. So why not here?

R56. We agree with the reviewer and have covered the issue of variable selection above (R4).

Lines 408-409 have already been criticised. Most of this paragraph needs rewriting. Where did the mixed forest conclusion come from – what in the results show this, if most of the forests are dominated by one species?

R57. We have now rewritten this section to make clear that the mixed forest argument refers to the mixing of forest stands with different attributes, which is what we have analysed in this paper (see R48 for a detailed response).

Line 439 states that clear-cutting is forbidden in Germany. But has it always been like this? When was clear-cutting banned in the areas studied? Can it have had any legacy effects?

R58. We have now rephrased the sentence to be more accurate. Managed plots in our study area include uneven-aged (selection) and even-aged systems. Even-aged systems employ shelterwood cuts in the regeneration phase. In the past, some conifer plots may have originated from clear-felling, but this practice is not applied anymore. In fact, beech forests have never been managed with clear-cuts and have always employed shelterwoods. This is because this method allows for the natural regeneration of beech stands and is therefore cost-effective. However, clear-cuts were applied in some regions in conifer forests until approx. 1980. This means that legacy effects may exist in conifer stands >40 years, but we believe that more recent treatments such as thinning have shaped the stand structure more significantly. In order to keep the manuscript focused and the discussion short, as requested by this reviewer, we avoid mentioning these potential legacy effects as we did not measure any variable related to that.

Lines 456-467. Which attributes were excluded, and was it only made based on VIFs and not on biological reasoning?

R59. As we specify in the methods and explain above (R29), all variables initially selected were expected to affect ecosystem services, and were therefore based on biological reasoning (i.e. we provide evidence of the effects of the different forest attributes on ecological functioning in the introduction, L. 98-111). We used VIFs afterwards to remove collinear predictors among these biologically-based variables. We further applied biological knowledge to retain the final 12 variables after the VIF procedure. In addition, we provide now Sup. Fig. 3 with the correlations of all variables explored initially.

Line 481. The measurement of horizontal heterogeneity is unclear to me. The plots were 1 ha in size, i.e. 10 000 square meters. So why were 25 400 m² (square meter) cells used per hectare? Then your measures are averages of partly overlapping cells. Or have I misunderstood something?

R60. Thank you for pointing out this mistake. There was a comma incorrectly placed in the method description before, which created the misunderstanding. We have now corrected this to explain that

we estimated the standard deviation of canopy height using 25 cells, each of 400 m² per hectare (i.e., per plot).

Line 563-567. The use of the Tukey's HSD tests assumes that observations are independent (I assume). So you need to adjust this somewhat to account for the non-independency of the correlation coefficients (see above). Alternatively, you could examine if each correlation (between X and Y) differs significantly after accounting for environmental differences. The hypothesis would be that they decrease significantly when the driving variable has been accounted for. You can use the test in Sokal & Rohlf (1981, pp. 583-91 according to my excel version, but I'm writing this on the train and cannot check it) outlining how to do this. I don't think this is very elegant, but it would be a way of checking the stability and consistency of the results.

R61. Thank you for the suggestion. We have explored this issue in detail and provided a full response to this point above in R42.

Supplementary materials. Most of my comments have been discussed above.

Line S258-. Is there no information WHEN harvesting was made. It could be important?

R62. We have followed reviewers' suggestions and use now biomass production data instead of harvest indices as a proxy of timber production. However, as the two measures (MAI and PAI) integrate over time, harvesting date is not an issue anymore.

Lind S266-. There is no information on which sources the list of edible fungi was made.

R63. We have now included this information in the methods section. Please, see detailed response above (R14).

Good luck! This MS contains a very fine idea, but deserves more work.

R64. We really appreciate the effort that Prof. Bengtsson put into reviewing our manuscript and we hope that we have addressed all his concerns. Thank you very much for your great contribution to improve this work!

Jan.bengtsson@slu.se

Reviewer #2 (Remarks to the Author):

GENERAL APPRECIATION

Managing forests for the supply of multiple ecosystem services is a major challenge for forest ecologists and managers around the globe since management practices necessary leads to trade-offs amongst many services making difficult the identification of optimal practices. The authors present a comprehensive study of 150 temperate forest plots (Germany) in which the effect of different drivers (3 management types, 4 environmental factors, and 11 forest stand attributes) on 16 ecosystem service proxies were assessed. Robust analyses and overall well documented supplementary information support their manuscript. They found that forest stand attributes are the most important drivers of final ecosystem services and of synergies between services, while environmental factors are the most important drivers of intermediate ecosystem services.

This study distinguished itself from previous studies on multiservice forest management in two main aspects:

- First, the number of ecosystem services evaluated, 16, is remarkable, and is well distributed among the four categories of services (provisioning, supporting, regulating, cultural). Most studies generally compare the supply of a limited number of services such as wood volume, carbon stocking, and some measure of biodiversity, well below the amount and range of services covered by the authors.

- Second, the authors provide a novel perspective in investigating the supply of ecosystem services by looking at their relationships with environmental factors and forest stand attributes. Previous studies have focused on the effect of given management practices on services. However, the effect of management practices varies with environmental conditions and with the variety of structural and compositional stand characteristics that they generate. Here, the authors propose to focus on these intermediate drivers, with the promising hypothesis that such analysis will provide a basis to conceive novel management approaches that would target the stand characteristics associated with greatest benefits amongst different services.

R65. Thank you very much for appreciating our work and for the general positive input.

Notwithstanding these advances, the study presents two important limitations that raises doubt of its suitability for publication in a high impact journal such as Nature Communications.

- First, while 150 forest plots were investigated, these plots did not cover a wide gradient of management intensities, as pointed out by the authors themselves. Additional forest plots, maybe from other European countries, to include intensive management such as clear cuts, as well as unmanaged coniferous plots, would have strengthened the study by making it more comprehensive and by potentially identifying the drivers of stronger trade-offs between services. Conclusions would have been of interest to a larger group of forest ecologists and managers confronted with strong trade-offs.

R66. We agree with the reviewer that a larger gradient of management intensities would probably have strengthened our conclusions. Unfortunately, we could not include additional forest plots from other countries or other projects in our analysis, as we are not aware of any other study that has collected the large amount of data analysed here (at least 14 ecosystem services and 12 forest features), in a coordinated sampling fashion and with the same methodological design across a wider range of management intensities.

Nevertheless, our forest plots do contain a range of different attributes and a gradient of management intensity typical for central Europe. For example, the range of most forest attributes measured in our forest plots is comparable to that of ~60% of the German Forest Inventory (e.g. for tree species richness and evenness, understory richness, beech regeneration, vertical heterogeneity, canopy cover, mean DBH and deadwood). We therefore feel that the results are relevant and of interest to a range of managers and ecologists. We also hope that the approach used here will inspire further work that could shed light on the effects of forest management on ecosystem services in other forest types. We also clearly acknowledge this limitation of our study in the main text (L. 435-452), to help the reader in interpreting where our conclusions may or may not apply.

- Second, the contribution of this study is limited by the absence of a final step that would propose creative or operational management approaches based on the authors' findings. While the study reports on the most important drivers of ecosystem services, the problem of identifying appropriate management practices that limits trade-offs amongst services remains. For example, the last sentence of the introduction reads: "Our results suggest that forest management practices can be targeted to reduce trade-offs and promote synergies between ecosystem services". The authors ought to provide at least a few examples of a set of targeted management practices that together achieve a reasonable multiservice forest management approach. Therefore, when the authors state

in their discussion: “our results clearly show the value of determining the effect of individual forest attributes on ecosystem service supply rather than only considering management types”, I have to disagree. The authors do demonstrate the effects of forest attributes on ecosystem services, but not the value of this demonstration.

R67. We have now rephrased the discussion to give more concrete examples of management. We argue that there are trade-offs between services because some are promoted by old growth, broad-leaved forests while others are higher in more open forests. This trade-off could be mitigated by creating gaps in mature beech forests. However, we acknowledge that we cannot really demonstrate that this would reduce trade-offs because we have not performed this experiment and do not include this type of forest in our analyses. Still, we believe this is an interesting point that is worthy of discussion. Therefore, in the updated version of the manuscript we speculate about which combination of forest attributes would promote the most services, and how this may inform managers and practitioners (L. 377-399, 462-473).

For example, we suggest that introducing disturbances in beech forests to create gaps in the canopy could potentially increase the supply of multiple services. We also suggest that management for multifunctional forests could be planned at the landscape scale, for example by creating or maintaining variation in species composition (conifer and broadleaf, oak and beech) and other forest attributes like canopy cover across the landscape. In addition, we have now included more examples of landscape scale management, such as the TRIAD approach for land-zoning allocation, as suggested by the reviewer. However, the levels of forest attributes shown to have positive or negative effects on ecosystem services can be achieved using different forestry techniques, and it is not the aim of this paper to discuss which techniques are most effective at promoting the desired forest attributes.

We hope that with these changes, the value of our approach to analysing forest attributes is clearer.

METHODOLOGICAL CONSIDERATIONS AND NEEDED CLARIFICATIONS

The importance of the temporal scale when measuring different services or evaluating the multifunctionality of forests is not well acknowledged. All services should be measured over the same time period, usually a rotation period. Since it is not the case, explanations as to why it is reasonable to do so should be provided. For example, C storage in trees is evaluated based on living tree volume while harvested timber is extrapolated from stumps, both quantities have therefore accumulated over different time periods.

R68. We agree with the reviewer and have now replaced the indicator of timber production by an indicator based on the rotation period, the mean annual increment –MAI– (or periodic annual increment –PAI– for unmanaged forests). However, each process in an ecosystem works on different time frames, from days to decades. Therefore, any study would just be able to take measurements in a particular time-period. We are confident that the approach of the Biodiversity Exploratories project is one of the most robust available, because all the data has been collected in the same plots and with standardised methodology appropriate to each particular variable. It would of course be desirable to have all of the services measured over the same time period and we can do this over the rotation period for some services. However, for others, such as cultural values or pest control, we lack the historical data necessary to calculate these over a rotation period. These are therefore snapshot measures, which we expect to relate to long-term mean values. We have now clarified the issue of time scale in the Supplementary methods (2^o paragraph).

The claim that forest attributes were the most important drivers of final ecosystem services while, environmental factors were the most important drivers of intermediate ecosystem services (in the abstract and result section) is too strong given that these observations are true only on average. Had the authors considered additional final and intermediate services, would these average results still hold?

R69. In response to this comment and others from reviewer one, we have now removed the classification of intermediate and final and analysed all ecosystem services together. We now focus on discussing differences between particular services, rather than between intermediate and final services. Our main results still hold, i.e. forest attributes are important predictor of most ecosystem services, although we also showed that some soil-related services are mostly explained by local environmental factors.

In determining the drivers of ecosystem services, the potential correlation between management type, for managed forests, and forest attributes, should be acknowledged. On L288, the authors report that the only associations of harvested timber with forest attributes (deadwood volume and canopy cover) in managed forests suggest that this ecosystem service affect forest attributes rather than being affected by them. Additional negative associations with forest attributes would have appear had intensive management plots been included in the analysis.

R70. We have now removed the factor of management type in the analysis to avoid misinterpretation of the data. We have also replaced the indicator of timber harvested; hence, the issue raised by the reviewer should be solved in the current version of the manuscript. It was always our intention to view forest attributes as being indicative of management, as management is the biggest driver of many of them. However, we acknowledge that including management categories in the analysis did not make this clear. We have therefore removed the categories and focus on the forest attributes as measures of management.

Supplementary figure 2 indicates a non-zero harvested timber in unmanaged plots. This is non-intuitive and requires clarification.

R71. As we have described in the methods, unmanaged plots have not been harvested for at least 50 years, but there could be stumps left from previous extractions that were accounted for as timber production following this method (Kahl et al. 2012). That is, however, not the common situation and most unmanaged plots have been set aside from production longer time ago: from Fischer et al. (2010) "In a few of the otherwise unmanaged forest plots in the Schwäbische Alb some treelets had been removed recently. As this intervention was very minor, we classified these plots as unmanaged".

We have now replaced the indicator of timber harvested to a measure of potential supply of this service (based on the mean annual increment), as all other services are quantified using measures of potential supply. This means that unmanaged forests are considered to potentially supply timber, even if this service is not used.

Fischer, M. et al. Implementing large-scale and long-term functional biodiversity research: The Biodiversity Exploratories. Basic Appl. Ecol. 11, 473–485 (2010).

Kahl, T. & Bauhus, J. An index of forest management intensity based on assessment of harvested tree volume, tree species composition and dead wood origin. Nat. Conserv. 7, 15–27 (2014).

Supplementary figure 2 indicates that pest control is much larger in managed coniferous plots than managed/unmanaged broadleaves. However, pest control has only been evaluated with regards to the bark beetles and their antagonists. Are these pests generalists or do they favour a

coniferous/broadleaf tree species? It is uncertain whether this result is biased by the choice of the pest species studied.

R72. Pest control in managed conifer forests was higher than in managed/unmanaged broad-leaved forests in our study, however, we are quite confident that this is not due to a methodological bias. We selected ambrosia beetles of the tribe Xyleborini because they are generalist species that are all found in conifer as well as broadleaved forests, with some showing preference either for broadleaved (e.g. *Trypodendron domesticum* (L. 1758)) or conifer (e.g. *Trypodendron lineatum* (Oliv. 1795)) trees. As we have no indication for different predation or parasitization rates in different species and we found high abundance of Xyleborini in conifer as well as broad-leaved forest stands (mean >>1000 Ind. per stand) we are confident that differences in the observed pest control are unbiased by the species group used.

We have included this additional information in the Supplementary Methods section.

Related with the previous comments, I wonder if the supply of certain services is de factor higher in broadleaves or coniferous forests. For example, floral aesthetic value also presents a large asymmetry between coniferous and broadleaves plots. A more solid comparison would require that managed and unmanaged broadleaves plots be compared together, and that managed coniferous plots be compared to unmanaged ones, which unfortunately, could not be done.

R73. The supply of floral aesthetic value is higher in broadleaved forests because the attractive plants are often typical forest specialists (eg. *Anemone nemorosa*). On the other hand, many wild edible species have higher values in managed coniferous forests because their canopies are more open, favouring a range of light demanding flowering plants. As the reviewer acknowledges it is impossible to separate the effect of management from the effect of the conifers because unmanaged conifer forests do not occur in the lowlands of central Europe. The dominant tree species is therefore an important aspect of management and managers do frequently favour non-native conifers in many parts of Europe, so the confounding effect of conifer presence and management is typical of many European forestry systems.

In the introduction, the line of arguments leading to your research objective needs to be strengthened. Emphasis should be put on one clear research objective that should be linked with one clear novel result. The introduction is long and the paragraphs are not clearly divided into the syntax rule of “one idea per paragraph”, leading to redundancy. Moreover, many vague terms or statements should be clarified, eg. “different aspects”, “broad types”, “large range” and will help tightening the introduction.

R74. We thank the reviewer for making this point. We have now rewritten and shortened the introduction and hope that it is now clearer about the research objectives.

The distinction between management type, management practice and harvesting method, should be clarified and use coherently throughout the text for the study and results to be appreciated by a broad readership. Additionally, the description of the analysed managed plots (“Managed plots in our study area include selection forestry and age class forestry, but not clear-cutting, as this practise is forbidden in Germany”) should be part of the main text and not simply in the Methods section, since “managed” refers to more intensive practices (clear-cut, plantation, etc) in many countries.

R75. We agree with the reviewer and have included this information in the main text of the paper. In addition, the main text has been rewritten throughout and we are confident that the message is now coherent and clear.

SMALLER COMMENTS

L62: In the abstract, the term “forest attributes” is vague. Indications should be given to the reader as to what kind of attributes your study is referring to: compositional, structural, functional?

R76. We agree with the reviewer and have now rewritten the abstract to address reviewer’s suggestion.

L62: It is important to add “stand-level” before “forest”.

R77. Done.

L63: “management types” should be defined. Again, it is vague when reading the abstract.

R78. We have now removed from the analyses the management types as a factor, so we have also removed it from the abstract.

L68-71: The last sentence of the abstract should be improved to emphasize the novelty of the work. Presently, the described results are not very surprising and the “particular forest attributes are required for some services” is quite generic.

R79. Thank you for the suggestion. We have now rewritten the abstract to address this point and provide some more specific results (e.g. “Our study suggests that managing forests to increase structural heterogeneity, maintain large trees and canopy gaps would promote the supply of multiple ecosystem services”).

L96: It is missing references about studies that did make distinction between different harvesting methods within a broad management type. For example Bradford & D’Amato. Front. Ecol. Env. 2012.

R80. Thank you for the suggestion. However, we have now removed this line of argumentation so this reference is no longer needed.

L110: This paragraph would benefit from adding an example of the effect of forest structure on a recreational or cultural services, since you point out the scarcity of such studies in the 1st paragraph.

R81. Thank you for the suggestion, we have now rewritten the introduction and added more examples, these are explained in L. 102-108.

L116: These are examples of the effect of harvest methods on forest structure, and not of forest structure on services as the first sentence announces. I suggest adding examples of services to which these structures are related.

R82. We agree with the reviewer that this point was not clear. We have now rewritten the paragraph to make the point that we don’t know the effect of particular forest structures on ES, as that is what we investigate in this study.

L118: Specify the kind of heterogeneity: age, diversity, vertical structure?

R83. We agree that this point was not clear. We have now rewritten the sentence, specifying that it is both vertical and horizontal heterogeneity (L. 103).

L123: There is an asymmetry between the examples for forest attributes and the ones for environmental conditions. I suggest adding examples for environmental conditions.

R84. Thank you for the suggestion, we have now rewritten the introduction and added more examples, these are included in L. 115-118.

L134: This first sentence of this paragraph should be: "There is limited information on how particular forest stand attributes affect relationships between multiple services". The other part of the sentence is one of the reasons of why this is the case, and it should be in a separate sentence as done for the other reasons.

R85. Thank you for the suggestion. We have rewritten the introduction incorporating this change.

L135: "a few pairs of services" should be replaced by "a limited number of services"

R86. Thank you for the suggestion. We have incorporated this change.

L138: This sentence is vague; it needs an example.

R87. Thank you for the suggestion. We have now rewritten most parts of the introduction to make it more specific.

L154: "wide gradient in management types" is in contraction with the three management types cited in the previous sentence.

R88. We agree with the reviewer and have now rewritten the sentence to be consistent throughout the text.

L164: I find the term "region" vague, given that it is its first occurrence in the text. Are you referring to bioclimatic region or location?

R89. Thank you for making this point. Region refers to each of the three locations (each one including 50 plots) where data was collected. We have followed reviewer suggestion and replace the term 'region' by 'location' throughout the text to make it clear.

L179: "forest types" should be replaced by "analysed forest plots"

R90. Thank you for the suggestion. We have incorporated this change.

L185: "effects" should be removed from "generally negative effects"

R91. Thank you for the suggestion. We have incorporated this change.

L189: "different aspects" is vague.

R92. We agree with the reviewer and have now rewritten the sentence to make the point clear.

L190: It should be specified that forest attributes on average are good predictors of final ecosystem services.

R93. We agree with the reviewer and have rewritten the sentence to make it clear.

L204: Add "stand-level" in front of forest attributes.

R94. Thank you for the suggestion. We have incorporated this change.

L208: Reference to figure 1b should follow reference to figure 1a. Sub-figures or result description should be interchanged.

R95. Thank you for the suggestion. We have now a single figure plot so this issue is solved.

L326: "services are higher" should be replaced with "services have higher value".

R96. Thank you for the suggestion. We have incorporated this change.

L367-379: The discussion about multifunctional landscape management should include references about functional zoning, such as the TRIAD approaches. Ex: Gustafsson et al. BioScience 2012, Messier et al. Forestry Chronicle 2009.

R97. Thank you for the suggestion, we have now included references to land-zoning allocation and the TRIAD approach in L.393-396.

Figure 1: Interchange sub-figures a and b. Moreover, "Management type" should be colored in red, instead of light brown.

R98. Thank you for the suggestion. We have now a single plot and have removed the factor 'management' so these issues are solved.

Figure 3: The pink for "Education" is hard to distinguish from the pink of "bird-watching".

R99. Thank you for the suggestion. We have now changed the colour-code to make it easier to distinguish.

Reviewers' Comments:

Reviewer #1:

Remarks to the Author:

Comments for the authors:

This is a substantial and well made revision of a very interesting manuscript. The authors have successfully answered most (but not all) of my previous criticisms and very well so. Many of the analyses have been remade och others checked in a way that is commendable.

I have gone through the text in detail, and I don't have many major issues remaining. However, in my view there are still some things that the authors need to take into account, and some contradictions between some results and the conclusions in the abstract and the text. These will be listed below. Before this, I want to state that this is a very fine manuscript, that I find both extremely interesting and overall the data are now very well and meticulously analysed.

Abstract:

Firstly, although it is acknowledged that the study is correlative (and cannot be anything else), the authors use the word "driver" in many cases. Far too many, in my strong opinion (and I don't like the use of the word driver at all, but I may be old-fashioned). According to the Oxford dictionary, "driver" means (among other things like driving a car) "A factor which CAUSES a particular phenomenon to happen or develop" (my capitals). I don't think this wording is appropriate (except when saying "potential driver" or similar) in a correlative study. Towards the end the authors say "good predictor" which I think is more appropriate. It is especially in the abstract that I find the word inappropriate. On line 66 the wording should be "... these (i.e. forest attributes) are the best predictors of most ecosystem services, and also good predictors explaining synergies and trade-offs ... Moreover, although forest attributes do explain more than environment for most ecosystem services (ES), the difference is overall not that large, and for single ES both environment, forest attributes and the joint effect of F & E explain most of the variation. So the Abstract (lines 65-68) needs to be reformulated to better reflect the study and its results. This can easily be done. (see also below)

Line 90. Can be rephrased as "... managed for other values like habitat conservation or recreation" to better reflect the multitude of management purposes.

Line 97. Change to "may decrease tree diversity" which is more clear.

Line 131. Change to "activity, will inevitably reduce soil C stocks".

Lines 150-156. Several variables, especially "tree diversity" is, as is discussed elsewhere, a bit problematic, since most of the stands are in essence almost monospecific stands with a few individuals of subordinate species, which means that tree diversity is not really a useful proxy for a diverse forest stand. Moreover, as is said elsewhere but should be clearly stated here, proportion of conifers (and also oak) are strongly and negatively correlated to beech cover ($r=-0.8$), which means that the interpretation of the variables are simple but need to be spelled out clearly from the start.

Line 199. Write "was usually considerable" rather than important. In my view.

General on line 199-211. The shared variance can be interpreted in many ways, and is not a straightforward thing and one should be vary about its interpretation. It is variation that cannot be attributed to one or the other factor, for several reasons. An obvious example is that it can be due to unmeasured factors that influence both F and E, but there are other possibilities like that the F and E are correlated in the data. The method is elegant and I do like it, but be vary when you attach a

meaning to the joint effect of F and E.

Line 221. In the Results section, the use of drivers is not appropriate, in my view. "Predictors" is more appropriate (but of course more boring) because there are no true causes among the results (and to be frank, you don't discuss and argue a lot for causation either).

Line 222-224 and Fig 2. I found the Figure 2 very difficult to interpret and non-intuitive. Then when I checked the Supplementary Figure 4 I realised what you had done. What is shown in the main figure is the CHANGE in correlations as E and E+F are accounted for, which means that panels 2 c and d are not standing on their own. For me (and probably most other readers) it would be more straightforward to show what's in Supp Fig 4 a and b in the main text, and put the CHANGES in the Supplementary material. I strongly suggest swapping the panels, as the interesting thing really is how the CORRELATION STRUCTURE changes, not the change as such. This would also show the major effect (that the authors now try to hide, unfortunately), namely that this analysis shows that it's mainly the environmental effects that change the correlations between ES, and NOT the forest attributes. This is shown in panel 2a in Fig 2. This is discussed on lines 234-238, but NOT in the Abstract (lines 66-67). THIS HAS TO BE CORRECTED, because the Abstract does not really reflect the results, but rather states that "forest attributes ... are ... the cause of synergies and trade-offs between services". This is NOT what the results show.

What I am demanding is that the good discussion on Lines 234-249 is reflected in the Abstract and the preceding paragraph (lines 221-232).

Line 276. Where are these results presented? Table 1 is not giving any details, and it's unclear what the results are (or I could not find them). The procedure is unclear from the text and not really obvious from the Methods section either. Since the method here seems to involve deleting variables (backwards simplification), more detail would be appropriate in the Suppl. materials (or if it's there, refer to it clearly in Table 1).

Lines 280-285. Is this the statistics in Suppl. Table 5? If so, give this information in the text.

Lines 285-301 approx. This discussion of the causes of different effects is sometimes puzzling and a bit unclear. For example, for the effect (?) of canopy cover on timber production is very difficult to know what is the probable cause and the effect. It is well known (from Scandinavia at least) that thinning (reducing cover) causes lower production (measured here as timber production). This suggests that it is the forest management (harvesting timber) that results in lower canopy cover on the plots rather than an effect of canopy cover on timber production. I may be wrong, but the cause-effect framework of classical experiments may not be appropriate in this (excellent) observational and correlative study. So can some of the effects rather be caused by harvesting beech than any effect of forest attributes, which actually are created by management?

Line 306-307. I don't agree at all. The results (Fig 2a, for example) show that it's the environment that is most important for explaining synergies and trade-offs, and that the additional effect of forest attributes is insignificant. The method is clever and elegant, but it seems that you don't want to acknowledge its results. YOU MUST ADJUST THE TEXT OVERALL (cf. above) SO THAT THIS RESULT IS CLEAR AND ACKNOWLEDGED, not brushed away.

Line 313. These results are not shown in the Suppl. materias, but important to understand what has been done. See also above!

Line 317. This figure is OK, but it should be noted that some ES were not significant in the full models.

Line 326. Should read "proxies were more related to" rather than responded etc.

Line 327. Add "factors than to the environment (Figure 1)." to make this clear.

Line 330. Alternatively (or additionally) belowground services are more dependent on slow soil processes and hence can be modified only on longer time scales.

Line 335. I would NOT state that shrub richness had "generally positive effects" on ES. There is in Fig. 3 2 ES that are positive and one that is negative to shrub richness, so this statement is clearly not true.

Line 346. This is not at all evident from Suppl. Fig. 1 (panel 4). Delete or reword!

Lines 351-52. Two "drivers" than need rewording.

Lines 350-375. This paragraph is too long and should be divided into three. Divide on line 355 and 367. Delete "of course" on line 360, and change 'driver' to 'explanation' on line 361. On line 370 'potential driver' is appropriate so it can be kept.

Line 377-379. Here it should be stated that conifer cover and beech cover are negatively correlated, and hence low values of conifer cover indicates beech cover.

New paragraph on line 389 would be natural.

Line 408-409. Shrub richness does NOT have overall positive effects on ES. Reword!

Line 416. It is unclear how this is evident in Suppl Fig 1.

Line 418-420. A question only. If tree C storage is dependent on one species of tree only (beech and spruce, respectively, in different stands) then any increase in evenness will decrease biomass (tree C storage) overall. So the particular stands in this study may make it very difficult to find effects of diversity at all.

Line 424-425. But it is more likely that tree (=trait) richness is the cause of structural heterogeneity than structural heterogeneity causing tree richness. So the preceding sentence suggesting that heterogeneity is important is not making sense. Or do I misinterpret something here?

Line 446. Add "broad" (categories) for clarity.

A good final paragraph!

Methods:

Section on Forest attributes.

It is still inconsistent to present the Forest attributes in the Methods text, but not the Ecosystem services. I suggest that both are briefly listed in the Methods, but the longer explanations of both are placed in the Supplementary Materials.

Explain DBH with words (line 526). That is "diameter at breast height".

Line 529. Give species in commonness order, and state which species are only found in a few plots.

Line 542. What variables were used for the Simpson index?

Overall on lines 550-563. The word "estimate" implies something approximate, a proxy. If you have real measures, you "measure", or alternatively "calculate" from 'exact' measures. So each "estimate" must be replaced by other words here. You DID really measure things on the plots.

Line 585-591. The standardisation gives equal weight to all ES regardless variation in the field. This needs to be stated because it may give overdue weight to variables with less variation between plots. Not much you can do about it, but it must be stated.

Lines 611-631. This is fine, but the description can be improved by clearly stating each step in the analysis.

Lines 631-639 can go to the Supplementary materials.

Finally, I note that you have assumed that all relationships are linear, and not at all considered whether you need to add variables that could account for humpshaped or biomodal relations. I can understand why you have not done so, as the analyses will be much more extensive in such a case, but it be good if you actually had checked this assumption, and provide the results of such a test in the supplementary materials. Or did you do it? It may be interesting to check this, if you really want to be sure that your results are valid.

I hope I haven't missed or misunderstood something important.

Jan.Bengtsson@slu.se

Reviewer #2:

Remarks to the Author:

Felipe-Lucia et al. have extensively revised their manuscript and have provided a clear and well-argued rebuttal.

The main changes to the manuscript with regards to my initial comments are:

(1) The authors have removed forest management types from their analysis. As such, they have conducted new statistical analysis that evaluate only the effects of forest attributes and that of environmental factors on ecosystem services. This is a major improvement as it removes the correlation between forest management and forest attributes and it answers one of the criticisms that I have raised.

(2) The authors now evaluate all ecosystem services individually and do not group services according to the final vs intermediate classification. Because ecosystem services respond differently to different drivers, treating them individually provides more specific and useful conclusions instead of general and averaged observations regarding groups of ES. This modification also answers one my concerns.

(3) The authors have replaced the harvested timber service by wood production, which they measure using mean or periodic annual increment. This new measure accounts for wood volume available that may or may not be harvested, and hence allows for a true comparison of this provisioning service between managed and unmanaged forests.

The authors have also improved the text by removing redundancies, rephrasing vague sentences and strengthen their line of arguments. Together these changes, I believe, enhance the value of the conclusions put forward by the manuscript since identified attributes can be more straightforwardly

linked to forest management practices.

Small comments

Revised Fig 2: I know that the addition of correlation matrices was suggested by Reviewer 1 and I certainly appreciate the added value of these matrices in the main text as opposed to the Supplementary Information but this new figure is very busy and requires simplification to be "reader friendly". I do not have specific suggestions, but maybe have 2a as a single figure and b,c,d as a color figure using identical colors, x-labels, and legend ?

Supplementary Fig 3 and 4: the font size of numbers should be decrease in order for negative values to fit within their "box".

Caption of Table 1: remove "Note that the overall...." since an equivalent sentence appear in the main text.

Manuscript NCOMMS-17-33179A

Felipe-Lucia et al. *Multiple forest attributes underpin the provision of multiple ecosystem services.*

Response to reviewers' comments:

Please, find below a detailed response to reviewers' comments. All changes referred in these comments have been highlighted in yellow in the main text, as requested by the Editor. We have also done a final edition of the text for clarity, which has also been marked in the main text.

Reviewers' comments:

Reviewer #1 (Remarks to the Author):

Comments for the authors:

This is a substantial and well made revision of a very interesting manuscript. The authors have successfully answered most (but not all) of my previous criticisms and very well so. Many of the analyses have been remade och others checked in a way that is commendable.

I have gone through the text in detail, and I don't have many major issues remaining. However, in my view there are still some things that the authors need to take into account, and some contradictions between some results and the conclusions in the abstract and the text. These will be listed below. Before this, I want to state that this is a very fine manuscript, that I find both extremely interesting and overall the data are now very well and meticulously analysed.

Thank you for the positive comments.

Abstract:

Firstly, although it is acknowledged that the study is correlative (and cannot be anything else), the authors use the word "driver" in many cases. Far too many, in my strong opinion (and I don't like the use of the word driver at all, but I may be old-fashioned). According to the Oxford dictionary, "driver" means (among other things like driving a car) "A factor which CAUSES a particular phenomenon to happen or develop" (my capitals). I don't think this wording is appropriate (except when saying "potential driver" or similar) in a correlative study. Towards the end the authors say "good predictor" which I think is more appropriate. It is especially in the abstract that I find the word inappropriate. On line 66 the wording should be "... these (i.e. forest attributes) are the best predictors of most ecosystem services, and also good predictors explaining synergies and trade-offs ... Moreover, although forest attributes do explain more than environment for most ecosystem services (ES), the difference is overall not that large, and for single ES both environment, forest attributes and the joint effect of F & E explain most of the variation. So the Abstract (lines 65-68) needs to be reformulated to better reflect the study and its results. This can easily be done. (see also below)

Thank you for the careful review of the terms used in this manuscript. We have now rewritten the abstract replacing the term 'driver' by reviewer's suggestions. We also acknowledge the second point of the reviewer regarding the role of environmental variables. We have now rephrased the sentence highlighting its importance "*Environmental factors also played an important a role, mostly in combination with forest attributes*", and we have also adapted the text to address reviewer's comments. Due to word limitations we have not been able to add further information to the abstract but we extensively discuss this point in the main text.

Line 90. Can be rephrased as "... managed for other values like habitat conservation or recreation" to better reflect the multitude of management purposes.
Thank you. We have followed this suggestion.

Line 97. Change to "may decrease tree diversity" which is more clear.
Done.

Line 131. Change to "activity, will inevitably reduce soil C stocks".
Done.

Lines 150-156. Several variables, especially "tree diversity" is, as is discussed elsewhere, a bit problematic, since most of the stands are in essence almost monospecific stands with a few individuals of subordinate species, which means that tree diversity is not really a useful proxy for a diverse forest stand. Moreover, as is said elsewhere but should be clearly stated here, proportion of conifers (and also oak) are strongly and negatively correlated to beech cover ($r=-0.8$), which means that the interpretation of the variables are simple but need to be spelled out clearly from the start.

Thank you for the suggestion to clarify this. We have now added this information to the paragraph where we shortly describe the analysis (L. 154-159). All these details are also discussed later in the text (eg. paragraph starting in L: 435 for tree diversity; L. 409-411 for the negative correlations between beech and conifer).

Line 199. Write "was usually considerable" rather than important. In my view.
Done.

General on line 199-211. The shared variance can be interpreted in many ways, and is not a straightforward thing and one should be vary about its interpretation. It is variation that cannot be attributed to one or the other factor, for several reasons. An obvious example is that it can be due to unmeasured factors that influence both F and E, but there are other possibilities like that the F and E are correlated in the data. The method is elegant and I do like it, but be vary when you attach a meaning to the joint effect of F and E.

We agree with the reviewer and we have been cautious when interpreting the results. As we discuss in L. 384-385, we cannot rule out shared responses to unmeasured factors as the cause of these correlations. In response to this comment, we have now added this point to the results, line 204-207.

Line 221. In the Results section, the use of drivers is not appropriate, in my view. "Predictors" is more appropriate (but of course more boring) because there are no true causes among the results (and to be frank, you don't discuss and argue a lot for

causation

either).

Thank you for the suggestion. We have now rephrased the subtitle of the results section to “*Predictors of synergies and trade-offs between ecosystem services*”, as the reviewer suggested, and we have been careful using the term ‘driver’, which has been replaced by other terms throughout the manuscript (eg. explaining, L. 236, 253, 254, etc.).

Line 222-224 and Fig 2. I found the Figure 2 very difficult to interpret and non-intuitive. Then when I checked the Supplementary Figure 4 I realised what you had done. What is shown in the main figure is the CHANGE in correlations as E and E+F are accounted for, which means that panels 2 c and d are not standing on their own. For me (and probably most other readers) it would be more straightforward to show what’s in Supp Fig 4 a and b in the main text, and put the CHANGES in the Supplementary material. I strongly suggest swapping the panels, as the interesting thing really is how the CORRELATION STRUCTURE changes, not the change as such. This would also show the major effect (that the authors now try to hide, unfortunately), namely that this analysis shows that it’s mainly the environmental effects that change the correlations between ES, and NOT the forest attributes. This is shown in panel 2a in Fig 2. This is discussed on lines 234-238, but NOT in the Abstract (lines 66-67). THIS HAS TO BE CORRECTED, because the Abstract does not really reflect the results, but rather states that “forest attributes ... are ... the cause of synergies and trade-offs between services”. This is NOT what the results show.

What I am demanding is that the good discussion on Lines 234-249 is reflected in the Abstract and the preceding paragraph (lines 221-232).

We included panels with the ‘change’ in the correlation because we thought that would make it easier for the reader to interpret the results, as the comparison between matrices is given straightaway. Figures in the Supplementary material were a bit ‘bulkier’ and that is the reason that we placed them there.

However, given that both reviewers did not agree with the panel composition, we have reorganized figure 2 to allow increasing the font size and the readability of the figure: new Figure 2 is previous Fig.2a and new Figure 3a-c is previous Fig.2 b-d, but swapping panels b and c for those in the Sup. Fig. 4.

In addition, we have now rewritten the abstract, and both paragraphs to accommodate the reviewer’s suggestion.

Line 276. Where are these results presented? Table 1 is not giving any details, and it’s unclear what the results are (or I could not find them). The procedure is unclear from the text and not really obvious from the Methods section either. Since the method here seems to involve deleting variables (backwards simplification), more detail would be appropriate in the Suppl. materials (or if it’s there, refer to it clearly in Table 1).

We are sorry that the results were not clearly explained. Table 1 shows the overall effect of forest attributes across ecosystem services, but only significant effects left after model simplification are shown. The complete effects prior to model simplification have now been included as Sup. Table 5, and we refer to it in the methods section of the main text (L. 672-673) and in Table 1.

Lines 280-285. Is this the statistics in Suppl. Table 5? If so, give this information in the text.

Thank you for pointing this out. We have now made explicit reference to both Sup. Table 6 (previous Sup. Table 5) and Sup. Table 7a and 8 in the text (note the new notation, L. 299).

Lines 285-301 approx. This discussion of the causes of different effects is sometimes puzzling and a bit unclear. For example, for the effect (?) of canopy cover on timber production is very difficult to know what is the probable cause and the effect. It is well known (from Scandinavia at least) that thinning (reducing cover) causes lower production (measured here as timber production). This suggests that it is the forest management (harvesting timber) that results in lower canopy cover on the plots rather than an effect of canopy cover on timber production. I may be wrong, but the cause-effect framework of classical experiments may not be appropriate in this (excellent) observational and correlative study. So can some of the effects rather be caused by harvesting beech than any effect of forest attributes, which actually are created by management?

This additional section of the results aims at solving some questions that arose in the previous revision. We made an effort to disentangle potential species specific effects, versus other forest attributes, on ecosystem services. We have been cautious in the language employed to avoid claims of causal effects here, but show positive or negative relationships (increases or decreases). To make it clearer, we have now rephrased this sentence to indicate the association between larger values of canopy cover and lower values of timber production, without going into the details of which is the cause and which is the consequence. We have additionally reworded several sentences of this paragraph, starting in L. 304-327.

Line 306-307. I don't agree at all. The results (Fig 2a, for example) show that it's the environment that is most important for explaining synergies and trade-offs, and that the additional effect of forest attributes is insignificant. The method is clever and elegant, but it seems that you don't want to acknowledge its results. **YOU MUST ADJUST THE TEXT OVERALL (cf. above) SO THAT THIS RESULT IS CLEAR AND ACKNOWLEDGED**, not brushed away.

In this sentence we want to emphasize the most important forest attributes that have an effect on ecosystem services. As we explain in the introduction and discuss later (L. 343-354), we aim to identify forest attributes because these can be managed, i.e. modified to increase or decrease targeted ecosystem services, while environmental variables cannot be modified (at least in the short-medium term). In addition, forest attributes are not completely unimportant in driving synergies and trade-offs: whilst it is true that overall they do not significantly change correlations (when the effects of forest attributes are removed after environmental effects, which is conservative because it allocates all shared variance to the environment), forest attributes do affect many individual correlations. If supporting services are removed, then overall the forest attributes do affect correlations and in addition, 28 individual correlations were significantly reduced in strength when forest attribute effects were removed. Environmental factors are therefore very important but forest attributes play a role too. We do not want to hide the effect of environmental factors, and indeed we acknowledge their importance in the results and discussion (L. 204-210, 235-237, 240-244, etc.). The purpose of this concluding sentence for the paragraph is to emphasize the existence of forest attributes that do have an effect on ecosystem services. However, we acknowledge that the previous sentence was unclear, and following reviewer suggestions, we have now rephrased the sentence to make it clear that this sentence refers to the comparison with other forest attributes, and replacing 'driving' by 'most important for explaining' (L.325-328).

Line 313. These results are not shown in the Suppl. materias, but important to understand what has been done. See also above!

We have now included the full effects on Sup. Table 5.

Line 317. This figure is OK, but it should be noted that some ES were not significant in the full models.

The ES that are significant in the full models are indicated with asterisks (*), we therefore hope it is clear which effects are not significant in full models.

Line 326. Should read "proxies were more related to" rather than responded etc.

Done

Line 327. Add "factors than to the environment (Figure 1)." to make this clear.

Done

Line 330. Alternatively (or additionally) belowground services are more dependent on slow soil processes and hence can be modified only on longer time scales.

Thank you for the suggestion. We have now incorporated this sentence.

Line 335. I would NOT state that shrub richness had "generally positive effects" on ES. There is in Fig. 3 2 ES that are positive and one that is negative to shrub richness, so this statement is clearly not true.

We have now rephrased this sentence to indicate that these results refer to the overall effects shown in Table 1 (L.356-358). In this overall analysis we analyse all services together and determine which variables had significantly positive or negative overall effects on the ecosystem services. Shrub cover did significantly increase services in this model because across all services, its positive effects outweighed the negative effects (despite the one negative effect in Fig. 4). We can therefore conclude that shrub richness had generally positive effects. We have now explained this more clearly in the text L. 358-363).

Line 346. This is not at all evident from Suppl. Fig. 1 (panel 4). Delete or reword!

We agree with the reviewer that this example was not clear and we have now used another example to make this point clear (L. 368-370).

Lines 351-52. Two "drivers" than need rewording.

We have rephrased the sentence, replacing 'driving' by 'explaining'.

Lines 350-375. This paragraph is too long and should be divided into three. Divide on line 355 and 367. Delete "of course" on line 360, and change 'driver' to 'explanation' on line 361. On line 370 'potential driver' is appropriate so it can be kept.

Thank you for all the suggestions, which have been all incorporated.

Line 377-379. Here it should be stated that conifer cover and beech cover are negatively correlated, and hence low values of conifer cover indicates beech cover.

We have included this sentence in L. 409-411.

New paragraph on line 389 would be natural.

Done

Line 408-409. Shrub richness does NOT have overall positive effects on ES. Reword!
These results refer to Table 1, where we show the overall positive effect of shrub richness on ES, see comment above.

Line 416. It is unclear how this is evident in Suppl Fig 1.
This can be observed by looking at panels 'tree richness' and 'tree evenness', where we observe the relatively small values of both richness and evenness, which indicates that there is a main tree species dominating the plot and a few other species that increase species richness but are not equally abundant. We have now specified the panels in Sup. Fig. 1 that help interpreting this (L.443), and was also pointed out in the new version of the introduction (L. 158-159).

Line 418-420. A question only. If tree C storage is dependent on one species of tree only (beech and spruce, respectively, in different stands) then any increase in evenness will decrease biomass (tree C storage) overall. So the particular stands in this study may make it very difficult to find effects of diversity at all.
We agree that the design of the Exploratories is not ideal for looking at tree diversity effects and that negative effects of evenness might well indicate that mixed stands perform less well than monocultures of beech or of spruce or pine.

Line 424-425. But it is more likely that tree (=trait) richness is the cause of structural heterogeneity than structural heterogeneity causing tree richness. So the preceding sentence suggesting that heterogeneity is important is not making sense. Or do I misinterpret something here?
The sentence was not very well phrased. We agree that heterogeneity should be increased by tree species richness (not the other way around). We wanted to say that if heterogeneity is correlated with tree species richness, then it might also explain some of the tree diversity effects seen in previous studies (this would mean that tree diversity had an indirect effect on functioning, through increasing structural heterogeneity). We have rephrased the two sentences to make this clearer, lines 455-458.

Line 446. Add "broad" (categories) for clarity.
Done

A good final paragraph!
Thank you!

Methods:

Section on Forest attributes.
It is still inconsistent to present the Forest attributes in the Methods text, but not the Ecosystem services. I suggest that both are briefly listed in the Methods, but the longer explanations of both are placed in the Supplementary Materials.
We thank the reviewer for the suggestion and have now moved the detailed explanations of both ecosystem services and forest attributes to the Sup. Materials while keeping a brief description of both in the main text.

Explain DBH with words (line 526). That is "diameter at breast height".
Done.

Line 529. Give species in commonness order, and state which species are only found in a few plots.

We have now sorted conifer species according to their total cover, from which we can see the species present in several (the former species) or in a few plots (the latter). The reader is then referred to the paper describing the setting of the study area (Fischer et al. 2010). According to the comment above, this information is now available in the Sup. Materials.

Line 542. What variables were used for the Simpson index?

The Simpson index uses the number of canopy layers that are effectively occupied by foliage and woody component. We have now included more information in the Sup. Methods section.

For completeness, we provide a more detailed explanation for the reviewers below: Our calculation of vertical heterogeneity is based on the 'effective number of layers', a measure that describes the heterogeneity of vertical stand structure. It is based on the inverse Simpson-Index and thereby quantifies the number of canopy layers that are effectively occupied by foliage and woody components. At each plot, we performed nine systematically distributed terrestrial laser scans, using a Faro Focus 3D (Faro Technologies Inc., Lake Marry, USA). The resulting three-dimensional point clouds were then converted into a voxel model with voxels of 20 cm side-length and were stratified into layers of one meter thickness. The inverse Simpson-Index was then used to quantify vertical stand structure by relating the number of voxels in each vertical layer to the total number of voxels; where p_i is the share of voxels in the i th vertical layer, relative to the total number of voxels. Thus, index values increase with a more even occupation of layers by foliage along the vertical axis as well as with increasing stand height.

See details in

Ehbrecht, M., Schall, P., Juchheim, J., Ammer, C., & Seidel, D. (2016). Effective number of layers: A new measure for quantifying three-dimensional stand structure based on sampling with terrestrial LiDAR. *Forest Ecology and Management*, 380, 212-223.

Ehbrecht, M., Schall, P., Ammer, C. & Seidel, D. (2017). Quantifying stand structural complexity and its relationship with forest management, tree species diversity and microclimate. *Agric. For. Meteorol.* 242, 1–9.

Overall on lines 550-563. The word "estimate" implies something approximate, a proxy. If you have real measures, you "measure", or alternatively "calculate" from 'exact' measures. So each "estimate" must be replaced by other words here. You DID really measure things on the plots.

We used the term 'estimate' because we did not count every single tree in the plot, we 'estimated' the total number of trees from an inventory in a subset of the plot (400 m² instead of 10000 m²). According to the comment above, this information is now available in the Sup. Materials.

Line 585-591. The standardisation gives equal weight to all ES regardless variation in the field. This needs to be stated because it may give overdue weight to variables with less variation between plots. Not much you can do about it, but it must be stated.

We agree with the reviewer on the effects of standardisation. Given that each ecosystem service is measured in different units, this is the only way to analyse all ecosystem services together and be able to compare the results across services. However, it is true that we may overestimate effects for services where we have sampled a small range of variation, compared to the maximum potential range. We have now added a sentence highlighting the issue (L.590-593). To address this we would need to define the theoretical range for each ecosystem service that we could expect in similar forest types. This is not well known for most services, so we cannot standardise variables by a theoretical range.

Lines 611-631. This is fine, but the description can be improved by clearly stating each step in the analysis.

We have adapted the text indicating clearly the order of the analyses in three steps (L. 615-619, L. 625, L. 630, L. 646, L. 647).

Lines 631-639 can go to the Supplementary materials.

We prefer to include the complete procedures of the statistical analyses in the main text to avoid misunderstandings in the procedures followed.

Finally, I note that you have assumed that all relationships are linear, and not at all considered whether you need to add variables that could account for humpshaped or bimodal relations. I can understand why you have not done so, as the analyses will be much more extensive in such a case, but it be good if you actually had checked this assumption, and provide the results of such a test in the supplementary materials. Or did you do it? It may be interesting to check this, if you really want to be sure that your results are valid.

Thank you for making this point. We tested for potential non-linear effects by visually inspecting all service x attribute relationships and we did not find evidence for non-linear relationships, although we didn't formally test for this. We have now specified this in L. 655-657.

I hope I haven't missed or misunderstood something important.

Jan.Bengtsson@slu.se

Reviewer #2 (Remarks to the Author):

Felipe-Lucia et al. have extensively revised their manuscript and have provided a clear and well-argued rebuttal.

The main changes to the manuscript with regards to my initial comments are:

(1) The authors have removed forest management types from their analysis. As such, they have conducted new statistical analysis that evaluate only the effects of forest attributes and that of environmental factors on ecosystem services. This is a major improvement as it removes the correlation between forest management and forest attributes and it answers one of the criticisms that I have raised.

(2) The authors now evaluate all ecosystem services individually and do not group services according to the final vs intermediate classification. Because ecosystem services respond differently to different drivers, treating them individually provides more specific and useful conclusions instead of general and averaged observations regarding groups of ES. This modification also answers one my concerns.

(3) The authors have replaced the harvested timber service by wood production, which they measure using mean or periodic annual increment. This new measure accounts for wood volume available that may or may not be harvested, and hence allows for a true comparison of this provisioning service between managed and unmanaged forests.

The authors have also improved the text by removing redundancies, rephrasing vague sentences and strengthen their line of arguments. Together these changes, I believe, enhance the value of the conclusions put forward by the manuscript since identified attributes can be more straightforwardly linked to forest management practices.

Thank you very much for your positive comments.

Small comments

Revised Fig 2: I know that the addition of correlation matrices was suggested by Reviewer 1 and I certainly appreciate the added value of these matrices in the main text as opposed to the Supplementary Information but this new figure is very busy and requires simplification to be “reader friendly”. I do not have specific suggestions, but maybe have 2a as a single figure and b,c,d as a colon figure using identical colors, x-labels, and legend ?

Thank you very for your suggestions to make the figure more digestible. We have now reorganized the panels following your and reviewer1’ suggestions, so matrices are separated in a column and all with the same colours and labels (new Figure 3).

Supplementary Fig 3 and 4: the font size of numbers should be decrease in order for negative values to fit within their “box”.

Done.

Caption of Table 1: remove “Note that the overall...” since an equivalent sentence appear in the main text.

Done.

Reviewers' Comments:

Reviewer #1:

Remarks to the Author:

Comments on Felipe Lucia, version 3

This MS has been very well revised. There is one inconsistency that I have discovered (and should have noted earlier, I admit) which needs to be corrected. There are also some small minor issues that need to be clarified and, possibly, modified accordingly. Also, before publication the editors need to check that the use of past and present tense in the text is consistent (some examples below). Otherwise, the authors have done a really good job.

Sorry for being late with this final review, but I have been on holidays, conferences and travelling on trains in Europe, and I wanted to read all the text carefully before submitting the review. Which did turn out to be important.

My main remaining problem is that the variable PEST CONTROL is not clearly explained or consistent. When "pest control" in the variable explanations in the Supplementary Materials (line S269-S282) is referring to other studies, it is defined as "Predator/prey ratios" that are considered indicating pest control (more predators/prey assumed to indicate better control of the pest). However, the authors use the inverse ratio (bark beetles/the sum of predators and parasitoids = prey/predator ratio) which means that pest control decreases with an increase in the ratio. Hence the results on pest control are wrongly interpreted, and there are rather positive correlations between pest control and mycorrhization, N-availability, C storage in soils, temperature regulation, C storage in trees, plant cultural value, and bird-watching potential. And negative with root decomposition, dung removal, P availability and harvested timber. The last is an interesting trade-off, as it suggests that production comes with a cost of pest control (rather than a synergy).

IF I am right (and I think I am), then only a few analyses need to be redone (I guess, because the real measure of "pest control" is the inverse of the one used and it shouldn't change most things except that the interpretations of ALL correlations, trade-offs and synergies must be changed accordingly. It however also means that the on average positive effect of conifer cover on ES most probably changes to an overall negative one (or at least to an overall no effect), so you need to re-do the analysis shown in Table 1 for conifer cover. It depends on the strength of the conifer cover - timber production relationship.

I am really sorry for not noticing this earlier. Alternatively you actually did use the Predator/Prey ratio, in which case the text explaining the variable is wrong. Only you know this.

A number of minor comments too:

On line 170, the text should read "... and, finally, timber production as ...".

On line 184-187, I find the wording a bit awkward. The four forest attributes did NOT have an "overall positive effect" but the effect was rather ON AVERAGE POSITIVE. This is especially clear as regards conifer cover, which had contrasting effects, but on average (using the method of z-score standardisation) positive effects (if I am right about the pest control definition above then this needs to be recalculated, of course).

Similarly, on line 294, the two "consistently" should be changed to "on average" (because the effects are often not consistently positive or negative in the analyses). Line 296, I would rather use "on

average" rather than "overall", and then add "... Table 1).

The authors are discussing contrasting effects on ES later on, but I think that this discussion may need to be changed if I am right above.

Line 282 (legend to Fig 3). Add "solid line" box.

Line 409. The "however" is not needed.

Line 474. Skip this "clearly" – the word is used a bit too often and fits better on line 484. It's not needed on line 474.

Line 480. Why not write "... intensive production forests where clearcutting is dominating"?

Line 481. Perhaps change to "... experimental studies that manipulate, e.g., forest attribute. The parentheses are not really needed.

I find the mix between present and past tense in the MS a bit inconsistent. For example, in the abstract most of the sentences are in the past tense ("investigated", "played" etc) but "Our results SHOW that forest attributes ARE (not 'were') the best predictors". Also on line 116, the text should read (indicators) "have been better explained" in accordance with line 117. And so on. In addition, "understory" is sometimes spelled "understory" – which I think is correct - and sometimes "understorey". Before publication, some editing by the journal will be needed.

Response to Reviewers

REVIEWERS' COMMENTS:

Reviewer #1 (Remarks to the Author):

Comments on Felipe Lucia, version 3

This MS has been very well revised. There is one inconsistency that I have discovered (and should have noted earlier, I admit) which needs to be corrected. There are also some small minor issues that need to be clarified and, possibly, modified accordingly. Also, before publication the editors need to check that the use of past and present tense in the text is consistent (some examples below). Otherwise, the authors have done a really good job.

Thank you again for your thorough review and your enthusiasm about this manuscript.

Sorry for being late with this final review, but I have been on holidays, conferences and travelling on trains in Europe, and I wanted to read all the text carefully before submitting the review. Which did turn out to be important.

My main remaining problem is that the variable PEST CONTROL is not clearly explained or consistent. When "pest control" in the variable explanations in the Supplementary Materials (line S269-S282) is referring to other studies, it is defined as "Predator/prey ratios" that are considered indicating pest control (more predators/prey assumed to indicate better control of the pest). However, the authors use the inverse ratio (bark beetles/the sum of predators and parasitoids = prey/predator ratio) which means that pest control decreases with an increase in the ratio. Hence the results on pest control are wrongly interpreted, and there are rather positive correlations between pest control and mycorrhization, N-availability, C storage in soils, temperature regulation, C storage in trees, plant cultural value, and bird-watching potential. And negative with root decomposition, dung removal, P availability and harvested timber. The last is an interesting trade-off, as it suggests that production comes with a cost of pest control (rather than a synergy).

IF I am right (and I think I am), then only a few analyses need to be redone (I guess, because the real measure of "pest control" is the inverse of the one used and it shouldn't change most things except that the interpretations of ALL correlations, trade-offs and synergies must be changed accordingly. It however also means that the on average positive effect of conifer cover on ES most probably changes to an overall negative one (or at least to an overall no effect), so you need to re-do the analysis shown in Table 1 for conifer cover. It depends on the strength of the conifer cover - timber production relationship.

I am really sorry for not noticing this earlier. Alternatively you actually did use the Predator/Prey ratio, in which case the text explaining the variable is wrong. Only you know this.

Thank you very much for pointing this out. We are very much sorry that this was not clarified before. As the reviewer indicates, it is the ratio between the Predator and the Prey which should be

considered as an indicator of Pest control, and that is indeed how we estimated it. We have now rewritten the methods section to make it clear how it was calculated: *“The ratio between the sum of predators and parasitoids vs. bark beetles was used as a proxy of pest control (i.e. (Predators + Parasitoids) / (Bark beetles))”*.

This means that the analyses and the interpretation of the results in the manuscript are correct and we do not need to re-do them. We really apologize for the lack of clarity in the previous versions.

A number of minor comments too:

On line 170, the text should read “... and, finally, timber production as ...”.

Done.

On line 184-187, I find the wording a bit awkward. The four forest attributes did NOT have an “overall positive effect” but the effect was rather ON AVERAGE POSITIVE. This is especially clear as regards conifer cover, which had contrasting effects, but on average (using the method of z-score standardisation) positive effects (if I am right about the pest control definition above then this needs to be recalculated, of course).

We apologise for the confusion. By "overall positive effect" we meant that a variable had a positive effect on average in the combined analysis. These results correspond to the analysis presented in Table 1, where we analyse the ‘overall or net effect’ of each forest attributes on all ecosystem services. This is estimated considering the whole pool of ecosystem services together, and using ecosystem service identity as a random factor. We have now changed mentions of "overall positive effects" to "net positive effects" or "positive effects on average". We hope that this makes it clearer that a feature doesn't have to have positive effects on every individual variable in order to have a positive effect in the combined model.

Similarly, on line 294, the two “consistently” should be changed to “on average” (because the effects are often not consistently positive or negative in the analyses). Line 296, I would rather use “on average” rather than “overall”, and then add “... Table 1).

We agree with the reviewer that in this case we cannot talk about ‘consistent’ results and rephrased it as ‘positive or negative net effect’ (current L. 265, and also in L. 279, 394, and Table 1).

L. 267: we replaced ‘overall’ by ‘on average’.

The authors are discussing contrasting effects on ES later on, but I think that this discussion may need to be changed if I am right above.

No changes needed, as the interpretation of the results is correct.

Line 282 (legend to Fig 3). Add “solid line” box.

Done.

Line 409. The “however” is not needed.

Done.

Line 474. Skip this “clearly” – the word is used a bit too often and fits better on line 484. It’s not

needed on line 474.

Done. We have replaced this by "still".

Line 480. Why not write "... intensive production forests where clearcutting is dominating"?

We now rephrased the sentence to "where clearcutting is frequent".

Line 481. Perhaps change to "... experimental studies that manipulate, e.g., forest attribute. The parentheses are not really needed.

Thank you for the suggestion! We now rephrased the sentence.

I find the mix between present and past tense in the MS a bit inconsistent. For example, in the abstract most of the sentences are in the past tense ("investigated", "played" etc) but "Our results SHOW that forest attributes ARE (not 'were') the best predictors". Also on line 116, the text should read (indicators) "have been better explained" in accordance with line 117. And so on. In addition, "understory" is sometimes spelled "understory" – which I think is correct - and sometimes "understorey". Before publication, some editing by the journal will be needed.

Thank you for checking the consistency of the tenses throughout the manuscript. We have now revised the text to ensure consistency. However, it is the journal policy to present the results of the abstract in the present tense. Also, that the last paragraph of the introduction should contain a brief summary of both the results and the conclusions written in present tense. In general, we have tried to use present tense and active voice throughout the text, as the journal policies advice. Nevertheless, we agree with the reviewer suggestion and used the past tense in L. 118, in concordance with the rest of the paragraph.

Finally, both terms 'understory' (US) and 'understorey' (UK) are correct, we have now checked the manuscript to ensure consistency and used the British spelling.